# VITRIX-CLIPIN: Enhancing Fine-Grained Visual Understanding in CLIP via Instruction Editing Data and Long Captions

**Ziteng Wang**[*]
The Chinese University of Hong Kong, Shenzhen
tommmmywangzt@gmail.com

**Siqi Yang**[*†]
Meituan
siqi.yang@uq.net.au

**Limeng Qiao**
Meituan
qiaolm@pku.edu.cn

**Lin Ma**[†]
Meituan
forest.linma@gmail.com

## Abstract

Despite the success of Vision-Language Models (VLMs) like CLIP in aligning vision and language, their proficiency in detailed, fine-grained visual comprehension remains a key challenge. We present CLIP-IN, a novel framework that bolsters CLIP's fine-grained perception through two core innovations. Firstly, we leverage instruction-editing datasets, originally designed for image manipulation, as a unique source of hard negative image-text pairs. Coupled with a symmetric hard negative contrastive loss, this enables the model to effectively distinguish subtle visual-semantic differences. Secondly, CLIP-IN incorporates long descriptive captions, utilizing rotary positional encodings to capture rich semantic context often missed by standard CLIP. Our experiments demonstrate that CLIP-IN achieves substantial gains on the MMVP benchmark and various fine-grained visual recognition tasks, without compromising robust zero-shot performance on broader classification and retrieval tasks. Critically, integrating CLIP-IN's visual representations into Multimodal Large Language Models significantly reduces visual hallucinations and enhances reasoning abilities. This work underscores the considerable potential of synergizing targeted, instruction-based contrastive learning with comprehensive descriptive information to elevate the fine-grained understanding of VLMs. Project is available here.

## 1  Introduction

The Contrastive Language-Image Pre-training (CLIP) model [30] has revolutionized vision-language representation learning by aligning visual and textual concepts within a unified embedding space. Trained on extensive web-scraped image-text pairs [34], CLIP exhibits remarkable zero-shot generalization across diverse tasks, serving as a foundational model for applications like image classification, cross-modal retrieval, object detection and segmention, and multimodal large language models (MLLMs) [31, 11, 20, 3].

Despite its success in high-level semantic understanding, CLIP demonstrates limitations in capturing fine-grained visual details such as color, quantity, and spatial relationships [16, 59, 50, 15, 23, 22, 49]. For instance, distinguishing between a "black cat with a yellow bow tie" and one with a "red bow tie"

---

[*]Equal contribution. Work done when Ziteng Wang worked as an intern with Meituan.
[†]Corresponding author.

39th Conference on Neural Information Processing Systems (NeurIPS 2025).

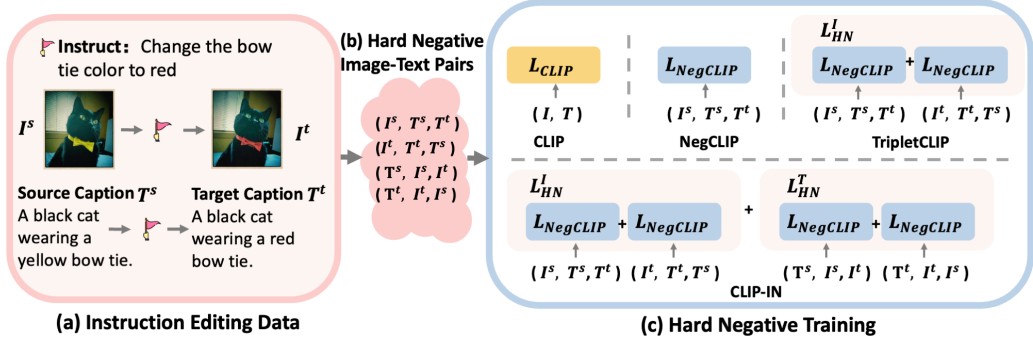

Figure 1: **Instruction Editing Data as Hard Negatives.** (a) We illustrate how instruction editing data provides challenging negative examples for CLIP. Given a source image and caption, an editing instruction leads to a target image and caption with subtle, fine-grained changes. The (source_image, target_caption) and (target_image, source_caption) pairs serve as hard negatives in (b), requiring the model to distinguish these nuanced visual-semantic differences. (c) We propose a symmetric hard negative contrastive loss to explicitly train the model to discern these subtle visual-semantic differences from both image-to-text and text-to-image perspectives.

can be challenging for CLIP when the overall scene context is similar (Figure 1). These limitations are inherited by MLLMs built upon CLIP's visual encoders, impacting their perceptual accuracy and contributing to issues like object hallucination [39, 46]. Benchmarks like Multimodal Visual Patterns (MMVP) [39] specifically highlight these "CLIP-blind spots" by presenting perceptually distinct image pairs that CLIP often confuses.

These shortcomings primarily arise from two aspects of CLIP's pre-training. First, the global image-text alignment objective can overlook subtle visual details and intricate inter-object relationships [16, 59, 23, 50]. Second, CLIP's text encoder, typically employing absolute positional embeddings with a fixed 77-token limit, restricts the effective utilization of longer, more descriptive captions that could provide richer supervision for fine-grained details [1, 55, 24, 58].

Existing approaches to enhance fine-grained understanding in CLIP have explored region-level contrastive learning [16, 59, 50], which often requires complex region proposals or additional supervision. Self-distillation methods [47, 23, 4, 22] aim to improve local-to-global consistency without explicit regional annotations but may suffer from weakened semantic grounding or be limited by the teacher model's capabilities. Hard negative mining strategies [53, 56, 27] focus on challenging the model with difficult negative examples, often generated by perturbing captions or using text-to-image synthesis, which may lack the necessary visual similarity or fine-grained control. Long-caption methods [1, 55, 24] extend CLIP's text processing capacity, but simply increasing the token limit does not guarantee improved fine-grained alignment or prevent performance degradation on short text inputs. This motivates the central question: How can we effectively and scalably enrich vision-language models like CLIP with robust fine-grained visual understanding by harnessing data sources that offer explicit, targeted supervision for subtle visual-semantic distinctions, while simultaneously leveraging the contextual richness of descriptive language?

In this work, we introduce **CLIP-IN** (CLIP with INstruction edit data and INformative long data), a novel framework that addresses this challenge by synergistically integrating instruction editing data and long descriptive captions. Our core innovation lies in repurposing instruction editing datasets, originally designed for image manipulation, as a valuable source of hard negative image-text pairs for contrastive learning. Datasets like UltraEdit [57] provide tuples of (source_image, source_caption, editing_instruction, target_image, target_caption), inherently offering hard negative examples with fine-grained differences in objects, attributes, and spatial relationships. This contrasts with synthetic hard negatives generated by text-to-image models, as seen in TripletCLIP [27], which often lack precise control and visual similarity to the source image. We propose a symmetric hard negative contrastive loss to explicitly train the model to discern these subtle visual-semantic differences from both image-to-text and text-to-image perspectives, implicitly learning fine-grained visual distinctions and their linguistic descriptions, unlike explicit regional contrastive learning [50, 15]. Additionally, We propose that long caption data is complementary to the instruction editing data. We leverage long descriptive captions to provide broader contextual richness. To enable existing CLIP

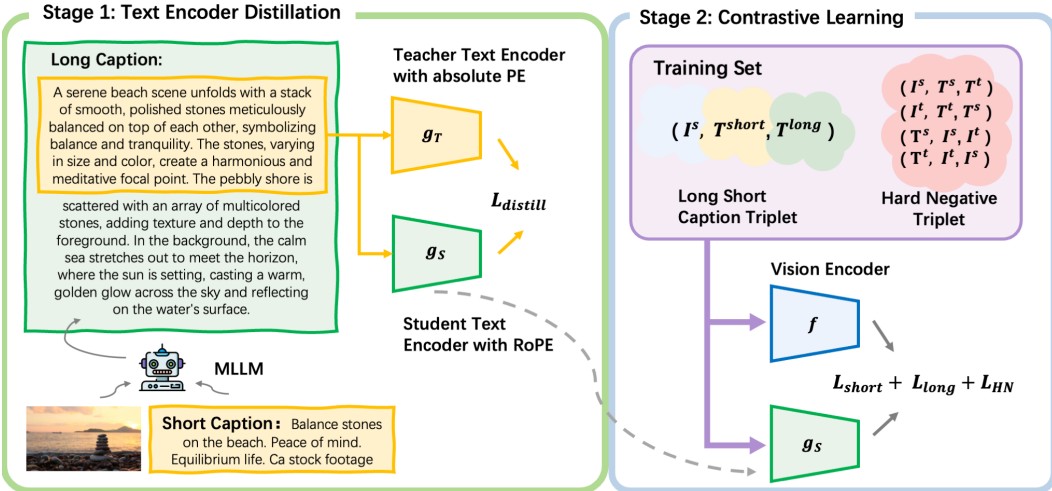

Figure 2: **CLIP-IN Framework Overview.** Stage1. We adapt the CLIP text encoder to process long captions using Rotary Positional Embeddings (RoPE) via knowledge distillation. Stage2. Our proposed framework, CLIP-IN, leverages two complementary data sources: instruction editing data and long descriptive captions. Instruction editing data excels at teaching the "where" and "how" of subtle visual details, while long captions provide the broader "what" and "why" of the scene, capturing complex relationships and contextual information.

models to effectively process long text, we adapt the text encoder by incorporating Rotary Positional Embeddings (RoPE) [36] through knowledge distillation from the pretrained clip text encoder model extended context.

By training on both instruction editing data and long captions, CLIP-IN aims to learn representations that are simultaneously semantically rich and visually grounded at a fine-grained level. Instruction editing data excels at teaching the "where" and "how" of subtle visual details, while long captions provide the broader "what" and "why" of the scene, capturing complex relationships and contextual information. This synergy enables CLIP-IN to outperform models trained on either data source alone.

Our primary contributions are: (1) A novel approach to utilize instruction-editing datasets as a rich source of hard negative image-text pairs for enhancing CLIP's fine-grained visual perception, extending the utility of these datasets beyond image generation tasks. (2) A synergistic training framework for CLIP that combines instruction-based image editing data and long descriptive captions to significantly improve its fine-grained visual-linguistic understanding by leveraging explicit and implicit supervision signals. (3) A methodology to adapt CLIP's text encoder for processing long captions by integrating Rotary Positional Embeddings (RoPE) via knowledge distillation, overcoming the inherent context length limitations. (4) Extensive experiments demonstrating consistent improvements on zero-shot visual benchmarks, fine-grained visual recognition tasks, and evaluations on MMVP and MLLM benchmarks.

## 2 Related Works

**Contrastive Language-Image Pre-training.** Contrastive Language-Image Pre-training (CLIP) [30] has demonstrated remarkable success in learning aligned visual and textual representations by contrasting positive image-text pairs against negative ones on a massive scale. Subsequent works like EVACLIP [37], MetaCLIP [51], SigLIP [54], and SigLIP2 [40] have further improved the performance and robustness of CLIP models through architectural modifications and training strategies.

**Fine-grained Understanding in CLIP.** Despite its effectiveness, standard CLIP often struggles with fine-grained visual understanding, including discerning subtle attributes, complex inter-object relationships, and precise object counts. This limitation stems, in part, from the typically concise nature of image captions used in pre-training, which may lack the detailed descriptions necessary for capturing such granular distinctions. To address this, several approaches have been proposed. GLIP [16], RegionCLIP [59], and FG-CLIP [50] leverage grounding data to explicitly align image regions with corresponding textual phrases, thereby enhancing the model's ability to understand local-

ized visual details. Complementary to this, methods like CLIPSefl [47], SILC [23], MaskEmb [4], and TIPS [22] focus on improving the consistency between global and local representations through self-supervised learning objectives, such as local-to-global correspondence learning via self-distillation and masked patch embeddings. Furthermore, DIVA [44] explores the use of generative feedback from text-to-image diffusion models to refine CLIP representations using only image data. However, these methods are primarily designed and evaluated on datasets with short captions, potentially limiting their effectiveness in scenarios requiring the processing of more detailed textual descriptions.

**Hard Negative Mining and Generation for CLIP.** Hard negatives, semantically distinct samples close in the embedding space, are crucial for learning discriminative features in contrastive learning. Prior works explored identifying hard negatives within datasets or generating synthetic negative captions [53, 56, 52, 35, 7, 6]. More recently, generating synthetic hard negative images has gained attention [28, 43, 33, 45]. For example, Peng *et al.* [28] synthesized images by rearranging segmented objects, potentially lacking real-world complexity and scalability. Sahin *et al.* [33] used inpainting for limited synthetic image generation, while Wang *et al.* [45] manipulated discrete visual tokens without generating corresponding negative texts. TripletCLIP [27], the most related work, generates hard negative image-text pairs using LLMs and text-to-image models for triplet loss. However, its generated images often lack control and deviate significantly, primarily enabling image-to-text hard negative learning. Our approach uniquely leverages large-scale instruction editing datasets like UltraEdit [57], which provide real images and localized visual changes, mitigating domain biases of purely generated data and yielding challenging hard negatives. This explicit grounding allows for symmetric hard negative losses addressing both image and text mismatches.

**Extending CLIP to Long Captions.** Standard CLIP models are typically limited in their text input length (e.g., 77 tokens), which can hinder their ability to process detailed and descriptive long captions. DCI [41] highlighted this limitation and, along with DOCCI [25], introduced datasets with dense, long captions for benchmarking. DreamLIP [58] proposed leveraging MLLM-generated long captions and dynamically sampling sub-captions to create multiple positive pairs, although it does not directly process the full long captions. Recent efforts [55, 15, 50] have extended CLIP's text capacity to 248 tokens by employing positional embedding interpolation and fine-tuning existing CLIP models. LoTLIP [48] incorporates long captions during the pre-training stage and introduces corner tokens to aggregate diverse textual information. TULIP [24] adopts a distillation approach, transferring the knowledge of a CLIP text encoder enhanced with relative positional embeddings (RoPE [36]). While these methods demonstrate progress in handling long captions, they often rely on large-scale datasets of long caption-image pairs (e.g., millions in [58, 55, 15, 24] and 1.6 billion in FG-CLIP [50]). Direct fine-tuning on such data can sometimes lead to performance degradation on tasks involving short text inputs, such as image classification. In this work, we aim to exploit the synergistic benefits of instruction-editing hard image-text pairs and long caption data to achieve improved fine-grained understanding without compromising performance on standard CLIP benchmarks.

## 3 Methodology

### 3.1 Preliminaries

**Contrastive Language-Image Pre-training (CLIP).** Given a batch of $N$ image-text pairs $(\mathcal{X}, \mathcal{Y}) = \{(x_i, y_i)\}_{i=1}^N$, CLIP aims to align semantically similar image-text pairs in a shared embedding space while separating dissimilar ones. The architecture consists of an image encoder $f(\cdot)$ and a text encoder $g(\cdot)$, both mapping their respective inputs to this common space. CLIP employs the InfoNCE loss [26] to achieve this alignment. For an image $x_i$ as the anchor, the image-to-text contrastive loss, $\mathcal{L}_{\text{I2T}}$, is defined as:

$$\mathcal{L}_{\text{I2T}}(\mathcal{X}, \mathcal{Y}) = -\frac{1}{N} \sum_{i=1}^N \log \frac{\exp(\langle f(x_i), g(y_i)\rangle/\tau)}{\sum_{k=1}^N \exp(\langle f(x_i), g(y_k)\rangle/\tau)} \,, \tag{1}$$

where $\langle \cdot, \cdot \rangle$ denotes the cosine similarity and $\tau$ is a temperature parameter. The overall CLIP loss, $\mathcal{L}_{\text{CLIP}}$, is a symmetric combination of the image-to-text and text-to-image contrastive losses:

$$\mathcal{L}_{\text{CLIP}} = \mathcal{L}_{\text{CL}}(\mathcal{X}, \mathcal{Y}) + \mathcal{L}_{\text{CL}}(\mathcal{Y}, \mathcal{X}) \,. \tag{2}$$

**Overview of CLIP-IN.** As depicted in Figure 2, our primary objective is to significantly enhance CLIP's capacity for discerning fine-grained visual details. To this end, we introduce CLIP-IN, a

**two-stage training framework** that leverages two complementary data modalities. First, **Instruction Editing Data** provides explicit supervision for learning localized visual transformations guided by precise textual instructions, thereby improving the model's sensitivity to subtle visual attributes and intricate relationships. Second, **Long Captions** offer rich semantic context and detailed descriptions of complex visual scenes, enabling a more comprehensive understanding beyond localized features. In the first stage of our framework, we adapt the text encoder to effectively process long captions by replacing absolute positional embeddings with RoPE and performing knowledge distillation. Subsequently, the second stage involves a joint contrastive learning process utilizing both instruction editing data and long captions to achieve the desired enhancement in fine-grained visual perception.

## 3.2 Instruction Editing Data as Novel Source for Hard Negative Training

Instruction editing data offers a unique form of supervision by explicitly linking textual instructions to precise visual modifications in real images, as exemplified in Figure 1. This structure is particularly valuable for generating challenging negative samples that target fine-grained visual distinctions.

**Formulating Hard Negatives from Instruction Editing Data.** Each instance in an instruction editing dataset is a tuple $(I^s, I^t, T^e, T^s, T^t)$, where $I^s$ is the source image, $I^t$ is the target (edited) image, $T^e$ is the editing instruction, and $T^s$ and $T^t$ are the corresponding source and target captions. From this data, we construct positive pairs $(I^s, T^s)$ and $(I^t, T^t)$, and critically, we define hard negative pairs as $(I^s, T^t)$ and $(I^t, T^s)$. These hard negatives are challenging because the image pairs $(I^s, I^t)$ and text pairs $(T^s, T^t)$ are minimally but semantically distinct due to the fine-grained edits described by $T^e$. This structured approach to generating hard negatives from real image edits provides a strong learning signal for enhancing fine-grained visual understanding. We utilize the recent UltraEdit dataset [57], comprising approximately 4 million high-quality instruction-based editing samples across diverse editing types and instructions, as our primary source of instruction editing data.

**Symmetric Hard Negative Contrastive Loss.** Inspired by prior work on hard negative mining in contrastive learning [53, 56], we formulate a symmetric hard negative contrastive loss. Given a hard triplet $(\mathcal{X}; \mathcal{Y}, \mathcal{Y}')$ where $\mathcal{X}$ is a set of anchor images, $\mathcal{Y}$ is the set of corresponding positive captions, and $\mathcal{Y}'$ is the set of hard negative captions, the hard negative contrastive loss for image-to-text alignment is:

$$\mathcal{L}_{\text{NegCL}}(\mathcal{X}; \mathcal{Y}, \mathcal{Y}') = -\frac{1}{N} \sum_{i=1}^{N} \log \frac{\exp(\langle f(x_i), g(y_i) \rangle / \tau)}{\sum_{k=1}^{N} \exp(\langle f(x_i), g(y_k) \rangle / \tau) + \sum_{m=1}^{N} \exp(\langle f(x_i), g(y'_m) \rangle / \tau)} .$$
$$(3)$$

From our instruction editing data, we derive two image-to-text hard triplets: $(\mathcal{X}^s, \mathcal{Y}^s, \mathcal{Y}^t)$ and $(\mathcal{X}^t, \mathcal{Y}^t, \mathcal{Y}^s)$, where $\mathcal{X}^s = \{I^s\}$, $\mathcal{Y}^s = \{T^s\}$, $\mathcal{X}^t = \{I^t\}$, and $\mathcal{Y}^t = \{T^t\}$. The image-to-text hard negative loss is then defined as:

$$\mathcal{L}_{\text{HN}}^{I} = \mathcal{L}_{\text{NegCL}}(\mathcal{X}^s, \mathcal{Y}^s, \mathcal{Y}^t) + \mathcal{L}_{\text{NegCL}}(\mathcal{X}^t, \mathcal{Y}^t, \mathcal{Y}^s) . \qquad (4)$$

Unlike methods like TripletCLIP [27] which struggled with text-to-image hard negative loss due to challenges in generating controlled negative images, our instruction editing data provides well-defined hard negatives in both modalities. Therefore, we propose a symmetric hard negative loss by also considering the text-to-image direction:

$$\mathcal{L}_{\text{HN}}^{T} = \mathcal{L}_{\text{NegCL}}(\mathcal{Y}^s, \mathcal{X}^s, \mathcal{X}^t) + \mathcal{L}_{\text{NegCL}}(\mathcal{Y}^t, \mathcal{X}^t, \mathcal{X}^s) . \qquad (5)$$

The final symmetric hard negative loss is the sum of both directional components:

$$\mathcal{L}_{\text{HN}} = \mathcal{L}_{\text{HN}}^{I} + \mathcal{L}_{\text{HN}}^{T} . \qquad (6)$$

This symmetric loss encourages the model to learn fine-grained visual-semantic distinctions from both image and text perspectives, leveraging the inherent structure of instruction editing data to create semantically meaningful and challenging negative examples. The illustration of the above hard negative training losses can be seen in Figure 1(b).

## 3.3 Synergizing Instructional Data with Semantically Rich Long Captions

**Generating Semantically Rich Long Captions.** To provide comprehensive semantic context that complements the fine-grained details learned from instruction editing data, we generate long,

descriptive captions for the images in our training set. We employ a state-of-the-art vision-language model, InternVL [3], to generate these detailed captions, which average around 300 tokens in length.

**Integrating Rotary Positional Encodings (RoPE).** Standard CLIP models with fixed-length absolute positional embeddings are limited in their ability to process long text sequences. To address this, we replace the absolute positional embeddings in the text encoder with Rotary Positional Encodings (RoPE) [36]. RoPE encodes absolute position through rotation matrices, enabling the self-attention mechanism to inherently consider relative positional information. This modification allows for greater flexibility in handling variable sequence lengths and improves extrapolation to longer inputs, which is crucial for processing our generated long captions.

**Knowledge Distillation for Relative Position Encoding.** To adapt the CLIP text encoder to utilize RoPE and effectively process both short and long captions while preserving its pre-trained knowledge, we employ a knowledge distillation strategy. This approach avoids the need for full retraining from scratch. Following insights from CLIP-KD [9] and TULIP [24], we initialize a student text encoder $g_S(\cdot)$ with the RoPE configuration and distill knowledge from a frozen teacher text encoder $g_T(\cdot)$ (the original CLIP text encoder). During this distillation phase, we exclusively use the generated long captions. For a long caption $y_{\text{long}}$ exceeding the teacher's context length $c_T$, we truncate it to $y_{\text{trunc}}$ of length $c_T$. The distillation loss, $\mathcal{L}_{\text{distill}}$, aims to align the embeddings produced by the student and teacher models for the truncated caption:

$$\mathcal{L}_{\text{distill}} = 1 - \frac{\langle g_T(y_{\text{trunc}}), g_S(y_{\text{trunc}}) \rangle}{\|g_T(y_{\text{trunc}})\| \cdot \|g_S(y_{\text{trunc}})\|} , \tag{7}$$

where $\langle \cdot, \cdot \rangle$ is the dot product and $\|\cdot\|$ denotes the L2 norm. After distillation, the student text encoder $g_S(\cdot)$ retains the capabilities of the teacher model within its original context length while gaining the ability to process longer sequences due to the integration of RoPE.

## 3.4 CLIP-IN Training Pipeline

Our CLIP-IN framework integrates the aforementioned components into a cohesive two-stage training pipeline designed to enhance fine-grained visual perception while maintaining general vision-language alignment capabilities. In the first stage, we adapt the text encoder by replacing absolute positional embeddings with RoPE and performing knowledge distillation using long captions, as described by Eq. 7. This step ensures the text encoder can handle variable-length sequences while retaining pre-trained knowledge. In the second stage, we jointly train the entire model using both instruction editing data and long caption data. We utilize the RoPE-enabled text encoder from the first stage. For the instruction editing data, we apply the symmetric hard negative contrastive loss $\mathcal{L}_{\text{HN}}$ (Eq. 6) to specifically target fine-grained visual understanding. For the long caption data, we employ the standard contrastive loss $\mathcal{L}_{\text{CL}}$ (Eq. 1) on the long captions paired with their corresponding images. Additionally, we also include the standard contrastive loss on the original short alt-text captions associated with the images to maintain performance on standard CLIP benchmarks. The overall training objective $\mathcal{L}$ is a weighted sum of these losses:

$$\mathcal{L} = \lambda_{\text{short}} \mathcal{L}_{\text{CL}}(\mathcal{X}, \mathcal{Y}_{\text{short}}) + \lambda_{\text{long}} \mathcal{L}_{\text{CL}}(\mathcal{X}, \mathcal{Y}_{\text{long}}) + \lambda_{\text{HN}} \mathcal{L}_{\text{HN}} , \tag{8}$$

where $\mathcal{Y}_{\text{short}}$ represents the original short alt-text captions, $\mathcal{Y}_{\text{long}}$ represents the generated long captions, and $\lambda_{\text{short}}$, $\lambda_{\text{long}}$, and $\lambda_{\text{HN}}$ are hyperparameters that balance the contribution of each loss term. The overview of this training pipeline is illustrated in Figure 2.

## 4 Experiments

### 4.1 Implementation Details

For hard negative training, we utilize the UltraEdit dataset [57], consisting of 4 million instruction-based image editing samples. For the long caption data, we employ InternVL2 [1] to generate detailed captions for approximately 18 million image-text pairs randomly sampled from a diverse set of datasets including CC3M, CC12M, COYO, and LAION [34]. All models are trained on 16 NVIDIA A100 GPUs with 80GB memory. We use an AdamW optimizer with a learning rate of 1e-4 and a weight decay of 0.05. The loss weights are set to $\lambda_{\text{short}} = 1.0$, $\lambda_{\text{long}} = 0.1$, and $\lambda_{\text{HN}} = 0.1$. We evaluate our framework by fine-tuning pre-trained state-of-the-art CLIP models. The global batch size for training with the ViT-L backbone is 16,384, and it is 4096 for the SigLIP2 models.

## 4.2 Zero-Shot Classification and Retrieval

**Datasets.** We assess the zero-shot image classification performance on the widely used ImageNet-1K dataset [5]. For zero-shot short-text retrieval, we use the Flickr30K dataset [29] and the MS-COCO dataset [18]. To evaluate zero-shot long-text retrieval capabilities, we utilize the DCI benchmark [41] and a 1K subset of the ShareGPT4V dataset [2], following previous works [50, 55].

**Results.** The results presented in Table 1 demonstrate the effectiveness of our CLIP-IN framework across various zero-shot evaluation tasks. Notably, our method consistently achieves competitive and often superior performance compared to strong baselines, including SigLIP2. For the SigLIP2 backbone, our approach shows clear improvements. With the ViT-SO/14 backbone at 224 resolution, our method achieves a slightly higher ImageNet-1K Top-1 accuracy (83.4%) compared to SigLIP2 (83.2%). More significantly, our method outperforms SigLIP2 in average short caption retrieval accuracy (78.8% vs. 76.4%) and average long caption retrieval accuracy (67.3% vs. 62.0%). A similar trend is observed with the larger ViT-SO/16 backbone at 384 resolution. While our method exhibits strong performance in long caption retrieval, it trails slightly behind FG-CLIP, which achieves an average of 81.8% compared to our 76.4% (with ViT-L/14 at 336 resolution). This discrepancy can be primarily attributed to the substantially larger long caption training dataset utilized by FG-CLIP (1 billion pairs) in contrast to our 18 million pairs. However, our ImageNet-1K classification accuracy of 77.0% significantly exceeds that of FG-CLIP (76.1%), indicating that our approach achieves a better balance between fine-grained retrieval and general image understanding despite using considerably less long caption data.

Table 1: Evaluation of zero-shot performance on various image benchmarks.

| Method | Backbone | Res | CLS IN-1K Top-1 | Short Caption Retrieval | | | | | Long Caption Retrieval | | | | |
| | | | | Flickr | | COCO | | | ShareGPT4V | | | DCI | |
| | | | | Avg | I→T | T→I | I→T | T→I | Avg | I→T | T→I | I→T | T→I |
|---|---|---|---|---|---|---|---|---|---|---|---|---|---|
| OpenAI CLIP [30] | ViT-L/14 | 224 | 75.5 | 60.7 | 85.2 | 64.9 | 56.3 | 36.5 | 64.3 | 84.2 | 83.7 | 45.3 | 44.0 |
| Ours | ViT-L/14 | 224 | **76.3** | **72.9** | **92.9** | **79.4** | **68.9** | **50.5** | **76.8** | **92.3** | **91.9** | **61.7** | **62.0** |
| OpenAI CLIP [30] | ViT-L/14 | 336 | 76.6 | 62.5 | 87.4 | 67.3 | 58.0 | 37.1 | 61.0 | 86.5 | 83.6 | 37.2 | 36.4 |
| EVA-CLIP [37] | ViT-L/14 | 336 | **80.4** | 69.8 | 89.2 | 77.9 | 64.2 | 47.9 | 69.0 | 91.5 | 89.4 | 47.2 | 47.8 |
| Long-CLIP [55] | ViT-L/14 | 336 | 73.5 | 68.8 | 90.0 | 76.2 | 62.8 | 46.3 | 72.0 | 95.8 | 95.6 | 44.2 | 52.5 |
| FineCLIP [15] | ViT-L/14 | 336 | 60.8 | - | - | - | - | - | 60.6 | 73.4 | 82.7 | 40.1 | 46.2 |
| FG-CLIP [50] | ViT-L/14 | 336 | 76.1 | **73.8** | 93.7 | **81.5** | **68.9** | 50.9 | **81.8** | **97.4** | **96.8** | **66.7** | **66.1** |
| Ours | ViT-L/14 | 336 | 77.0 | 73.1 | **93.8** | 79.3 | 68.2 | **51.1** | 76.4 | 93.5 | 91.6 | 58.4 | 61.9 |
| DFN-H [8] | ViT-H/14 | 224 | 83.4 | 74.8 | 92.8 | 80.1 | 72.3 | 53.9 | 79.8 | 92.5 | 90.3 | 68.7 | 67.5 |
| Ours | ViT-H/14 | 224 | 83.4 | 75.6 | 93.0 | 80.8 | 73.6 | 54.8 | 81.5 | 93.8 | 92.4 | 70.5 | 69.1 |
| DFN-H [8] | ViT-H/14 | 378 | **84.4** | 75.9 | 94.0 | 82.0 | 71.9 | 55.6 | 82.3 | 93.9 | 92.5 | 71.6 | 71.0 |
| Ours | ViT-H/14 | 378 | 84.1 | **76.8** | **94.6** | **82.2** | **74.0** | **56.4** | **83.5** | **95.4** | **93.9** | **72.7** | **71.9** |
| SigLIP2 [40] | ViT-SO/14 | 224 | 83.2 | 76.4 | 94.6 | 84.3 | 71.5 | 55.1 | 62.0 | 76.4 | 76.2 | 45.4 | 50.0 |
| Ours | ViT-SO/14 | 224 | 83.4 | 78.8 | 94.9 | 85.1 | 76.2 | 58.9 | **67.3** | **81.5** | **80.7** | **52.5** | **54.4** |
| SigLIP2 [40] | ViT-SO/16 | 384 | **84.1** | 77.1 | 95.9 | 85.3 | 71.2 | 56.0 | 59.1 | 70.7 | 72.8 | 43.4 | 49.6 |
| Ours | ViT-SO/16 | 384 | 83.7 | **79.6** | **96.4** | **85.6** | **76.5** | **59.8** | 64.0 | 77.7 | 76.4 | 50.0 | 51.7 |

## 4.3 Fine-Grained Visual Perception Evaluation

**Datasets.** We first evaluate fine-grained visual perception on the Multimodal Visual Patterns (MMVP) benchmark, which is specifically designed to probe the weaknesses of vision-language models in perceiving subtle visual differences. To evaluate the ability to understand the composition of images, we further evaluate on the Winnoground [38], SugarCrepe [14],SPEC [28] and ARO [53].

**Results.** As detailed in Table 2, our CLIP-IN model demonstrates a substantial improvement in fine-grained visual perception on the MMVP benchmark. With a ViT-L/14 backbone, our approach elevates the average accuracy from 18.5% to 30.4% (+11.9pp), with particularly striking gains in recognizing object State/Condition (+33.3pp) and in Feature Detection (+20.0pp). Furthermore, our SigLIP2-based variant not only surpasses the original's average accuracy (36.3% vs. 35.6%) but also shows marked improvements in complex categories such as Positional Context (+13.3pp)

Table 2: Performance of CLIP based models on various visual patterns of MMVP-VLM benchmark. Symbols for visual patterns as ([39]) are inherited: ⊘: Orientation and Direction, Q: Presence of Specific Features, ⟳: State and Condition, ↕: Quantity and Count, , ♟: Positional and Relational Context, ☻: Color and Appearance, ⚙: Structural and Physical Characteristics, A: Texts, ▣: Viewpoint and Perspective.

| Method | Backbone | Res | ⊘ | Q | ⟳ | ↕ | ♟ | ☻ | ⚙ | A | ▣ | Avg |
|---|---|---|---|---|---|---|---|---|---|---|---|---|
| OpenAI CLIP [30] | ViT-L/14 | 224 | 6.7 | 13.3 | 20.0 | 20.0 | 13.3 | 53.3 | 20.0 | 6.7 | 13.3 | 18.5 |
| Ours | ViT-L/14 | 224 | 6.7 | **33.3** | **53.3** | 20.0 | 13.3 | **60.0** | **33.3** | **26.7** | **26.7** | **30.4** |
| OpenAI CLIP [30] | ViT-L/14 | 336 | 0.0 | **20.0** | 40.0 | **20.0** | 6.7 | 20.0 | 33.3 | 6.7 | **40.0** | 20.0 |
| DIVA [44] | ViT-L/14 | 336 | **26.7** | 20.0 | 33.3 | 13.3 | **13.3** | 46.7 | 26.7 | 6.7 | **40.0** | 25.2 |
| Ours | ViT-L/14 | 336 | 13.3 | 13.3 | **46.7** | 13.3 | **13.3** | 53.3 | 33.3 | 20.0 | 28.3 | 26.1 |
| DFN [8] | ViT-H/14 | 224 | 20.0 | 26.7 | 73.3 | 26.7 | 26.7 | 66.7 | **46.7** | 20.0 | 53.3 | 39.9 |
| Ours | ViT-H/14 | 224 | 20.0 | 26.7 | 73.3 | 26.7 | **33.3** | 66.7 | **46.7** | 26.7 | 53.3 | **41.5** |
| DFN [8] | ViT-H/14 | 378 | 13.3 | 20.0 | 53.3 | **33.3** | 26.7 | 66.7 | 40.0 | 20.0 | 40.0 | 34.8 |
| Ours | ViT-H/14 | 378 | 13.3 | 20.0 | 60.0 | **33.3** | 26.7 | 66.7 | 40.0 | 20.0 | 46.7 | 36.3 |
| SigLIP2 [40] | ViT-SO/14 | 224 | 13.3 | **20.0** | 60.0 | 26.7 | 6.7 | **80.0** | 53.3 | 20.0 | 40.0 | 35.6 |
| Ours | ViT-SO/14 | 224 | 13.3 | 13.3 | 60.0 | 26.7 | **20.0** | 80.0 | 46.7 | 13.3 | 53.3 | **36.3** |
| SigLIP2 [40] | ViT-SO/16 | 384 | 13.3 | 20.0 | 46.7 | **40.0** | 20.0 | 73.3 | **53.3** | 6.7 | 46.7 | 35.6 |
| Ours | ViT-SO/16 | 384 | 13.3 | **20.0** | 60.0 | 33.3 | 26.7 | 66.7 | 40.0 | **20.0** | 46.7 | **36.3** |

Table 3: Evaluation on compositional reasoning benchmarks.

| Method | Backbone | Res | ARO | | | MMVP | Winoground | | | | SugarCrepe | SPEC | | |
|---|---|---|---|---|---|---|---|---|---|---|---|---|---|---|
| | | | Avg | relation | attribute | | Avg | text | image | group | | Avg | T->I | I->T |
| OpenAI CLIP [30] | ViT-L/14 | 224 | 58.9 | 59.3 | 58.5 | 18.5 | 15.9 | 28.3 | 10.5 | 8.8 | 75.6 | 32.3 | 33.2 | 31.3 |
| Ours | ViT-L/14 | 224 | **64.4** | **64.3** | **64.4** | 30.4 | 17.1 | 28.0 | **13.8** | **9.5** | **77.5** | **36.3** | **37.6** | 35.0 |
| OpenAI CLIP [30] | ViT-L/14 | 336 | 61.0 | 60.1 | 61.9 | 20.0 | 15.4 | 28.3 | 10.5 | 7.5 | 74.8 | 32.1 | 32.8 | 31.1 |
| Ours | ViT-L/14 | 336 | 60.7 | 58.1 | 63.2 | 26.1 | **18.1** | **33.0** | 11.8 | **9.5** | 77.2 | 35.2 | 35.1 | **35.2** |
| SigLIP2 [40] | ViT-SO/14 | 224 | 49.7 | 49.0 | 50.4 | 35.6 | 6.9 | 9.0 | 9.3 | 2.5 | 49.5 | 27.3 | 27.4 | 27.2 |
| Ours | ViT-SO/14 | 224 | 50.7 | 49.5 | 51.9 | **36.3** | 8.5 | 14.3 | 7.5 | 3.8 | 50.5 | 30.5 | 30.6 | 30.4 |
| SigLIP2 [40] | ViT-SO/16 | 384 | 48.9 | 47.3 | 50.5 | 35.6 | 6.7 | 9.3 | 8.5 | 2.3 | 50.9 | 27.5 | 27.6 | 27.5 |
| Ours | ViT-SO/16 | 384 | 50.5 | 50.9 | 50.0 | **36.3** | 7.0 | 13.5 | 5.5 | 2.0 | 51.7 | 30.5 | 30.2 | 30.8 |

and Viewpoint (+13.3pp). These results affirm that our training strategy effectively addresses the well-known limitations of standard CLIP models in discerning subtle visual patterns.

Building upon this enhanced perceptual acuity, CLIP-IN also exhibits superior compositional reasoning, consistently outperforming strong baselines across a suite of challenging benchmarks, as shown in Table 3. On ARO, which evaluates attribute, relation, and compositional understanding, our model achieves 58.9 avg vs. OpenAI CLIP's 58.4 (+0.5pp) at 224 resolution and 60.5 vs. 59.7 (+0.8pp) at 336 resolution, with notable gains in relation (64.3 vs. 64.3) and attribute (58.5 vs. 58.4), indicating improved sensitivity to subtle visual details. On Winoground, a challenging minimal-pair benchmark, CLIP-IN significantly improves text-to-image accuracy from 28.3% to 33.0% (+4.7pp), image-to-text from 10.5% to 11.8% (+1.3pp), and group accuracy from 7.5% to 9.5% (+2.0pp), highlighting its enhanced capability to ground small linguistic changes in distinct images. On SugarCrepe, designed to evaluate mitigative reasoning through hard negatives, our model achieves 79.4% average accuracy vs. CLIP's 73.8% (+5.6pp), reflecting robustness in attribute binding and relational reasoning. Finally, on SPEC, which focuses on precise spatial and compositional understanding, CLIP-IN surpasses CLIP by 34.8% vs. 32.0% (+2.8pp) and Siglip2 by 30.5% vs. 27.5% (+3.0pp), with gains in both T→I (35.1% vs. 32.8%) and I→T (35.2% vs. 31.1%) directions, confirming stronger cross-modal alignment under synthetic but precise conditions. These consistent improvements across diverse architectures (ViT-L and so400m) and tasks validate that our method—combining instruction editing and long caption training—effectively enhances compositional reasoning in vision-language models.

## 4.4 Evaluation on MLLM

We adopt LLaVA-1.5 [19] as the baseline framework to explore the potential of our proposed visual encoders in MLLM.

**Datasets.** We train our model with the same setting in LLaVA-1.5 and evaluate model performance on various multimodal understanding benchmarks (*i.e.,* MMVP [39], POPE [17], MME-Perception [10], MMBench [21], MMBench-CN [21], LLaVA-Bench-in-the-Wild [20].

**Results.** Table 4 presents the performance of LLaVA-1.5 when equipped with different visual backbones. We compare the performance of our CLIP-IN enhanced visual encoder against the original OpenAI CLIP backbone and DIVA [44], a recent method aimed at improving CLIP's visual representations. The results demonstrate that using our CLIP-IN visual backbone leads to significant improvements across several benchmarks, particularly on MMVP, MME, and both English and Chinese versions of MMBench, indicating enhanced fine-grained perception and overall multimodal understanding capabilities.

Table 4: Performance gains achieved by our enhanced CLIP visual backbone for MLLM. All methods use OpenAI ViT-L/14 at 336×336 resolution as pretrained backbone.

| Method | ViT | LLM | MMVP | POPE | | | MME | MMBench | | LLaVA-Wild |
| | | | | rand | pop | adv | | en | cn | |
|---|---|---|---|---|---|---|---|---|---|---|
| | OpenAI CLIP [30] | | 24.7 | 87.3 | 86.1 | 84.2 | 1510.7 | 64.3 | 58.3 | 65.4 |
| LLaVA-1.5 [19] | DIVA [44] | Vicuna-7B | **31.3** | 87.9 | 87.0 | 84.6 | 1500.6 | 66.4 | 60.6 | 66.3 |
| | Ours | | 28.0 | **88.5** | **87.2** | **85.2** | **1709.0** | **72.9** | **70.3** | **68.5** |

Table 5: Comparisons on synthetic data and data scale.

| Method | Training Data | IN-1K | Retrieval | Flickr | | COCO | |
| | | | Avg | T→I | I→T | T→I | I→T |
|---|---|---|---|---|---|---|---|
| Ours | TripletData 1.4M | **76.3** | 70.58 | 77.6 | 92.0 | 47.9 | 64.8 |
| | UltraEdit 1.4M | 75.8 | **72.9** | 78.3 | **92.9** | **51.1** | **69.2** |
| | UltraEdit 4M | **76.3** | **72.9** | **79.4** | **92.9** | 50.5 | 68.9 |

## 4.5 Ablation Studies

**Data Scale and Source Analysis.** To demonstrate the effectiveness of instruction editing data, we compare our approach with training using TripletData [27], a method that generates synthetic hard negative image-text pairs based on text-to-image generation model. Since TripletData provides 1.4 million samples, we also train our model using a randomly sampled subset of 1.4 million examples from UltraEdit for a fair comparison.Using a balanced subset of 1.4 million samples from UltraEdit and TripletData, UltraEdit showed a 2.32% increase in average retrieval accuracy (72.9% vs. 70.58%). On ImageNet-1K, UltraEdit's accuracy was 75.8%, slightly less than TripletData's 76.3%. Expanding UltraEdit to 4 million samples improved ImageNet-1K accuracy to match TripletData at 76.3%, while retrieval accuracy stayed at 72.9%. This indicates high-quality instruction editing data significantly boosts fine-grained retrieval and improves image classification with more data.

**Impact of Instruction Editing Data and Long Caption Data.** Tables 6 and 7 reveal the complementary nature of our two data sources. Instruction editing data alone improves retrieval accuracy from 62.5% to 72.0% (+9.5pp), confirming that hard negatives enhance fine-grained discrimination. Long caption data similarly boosts retrieval to 71.0% (+8.5pp) and ImageNet-1K accuracy to 76.8% (+0.2pp). Most importantly, their combination yields the highest retrieval performance (73.1%), best classification accuracy (77.0%), and significant MLLM improvements. While each data source alone decreases MMVP performance in LLaVA-1.5, their combination substantially improves it (+3.3pp), demonstrating how these complementary signals effectively address CLIP's fine-grained perception limitations.

**Impact of Rotary Positional Embeddings (RoPE).** To isolate the benefit of our RoPE-based text encoder distillation, we compare it against a strong baseline that uses absolute positional encoding

Table 6: Ablation studies on the contribution of different components in CLIP-IN.We evaluate the impact of instruction editing data and long captions, both individually and in combination, using OpenAI ViT-L/14 at 336×336 resolution as pretrained model.

| Method | InstructData | LongData | IN-1K | Retrieval Avg | Flickr T→I | Flickr I→T | COCO T→I | COCO I→T |
|---|---|---|---|---|---|---|---|---|
| OpenAI ViT-L/14 [30] | - | - | 76.6 | 62.5 | 67.3 | 87.4 | 37.1 | 58.0 |
| Ours | ✓ | | 76.6 | 72.0 | 78.5 | 93.0 | 49.7 | 66.6 |
| | | ✓ | 76.8 | 71.0 | 78.3 | 92.6 | 48.4 | 64.8 |
| | ✓ | ✓ | **77.0** | **73.1** | **79.3** | **93.8** | **51.1** | **68.2** |

Table 7: Ablation studies on the contribution of different components in CLIP-IN on the MLLM benchmarks.

| Method | InstructData | LongData | MMVP | POPE rand | POPE pop | POPE adv | MME | MMBench en | MMBench cn | LLaVA-Wild |
|---|---|---|---|---|---|---|---|---|---|---|
| LLaVA-1.5 [19] | - | - | 24.7 | 87.3 | 86.1 | 84.2 | 1510 | 64.3 | 58.3 | 65.4 |
| Ours | ✓ | | 22.1 | 85.4 | 84.4 | 83.8 | 1679 | 68.4 | **72.1** | 67.0 |
| | | ✓ | 20.9 | 85.6 | 84.8 | 82.5 | 1646 | 67.1 | 71.6 | 65.9 |
| | ✓ | ✓ | **28.0** | **88.5** | **87.2** | **85.2** | **1709** | **72.9** | 70.3 | **68.5** |

interpolation, as popularized by LongCLIP [55]. We initialize our second stage of training with LongCLIP, a text encoder pre-trained using positional interpolation and compare it to our proposed RoPE-based model. All other settings are identical. We use the ViT-L/14@224 model for this ablation. The results in Table 8 show that our RoPE-based approach is a more effective method for extending the text encoder's context length. It significantly outperforms the interpolation baseline on general classification (IN-1K: +6.2%), short-text retrieval (+3.5%).

Table 8: Ablation studies on the impact of Rotary Positional Embeddings (RoPE).

| Method | CLS IN-1K Top-1 | Short Caption Retrieval Avg | Flickr I→T | Flickr T→I | COCO I→T | COCO T→I | Long Caption Retrieval Avg | ShareGPT4V I→T | ShareGPT4V T→I | DCI I→T | DCI T→I |
|---|---|---|---|---|---|---|---|---|---|---|---|
| Ours(w LongCLIP) | 70.1 | 69.4 | 89.1 | 76.8 | 64.7 | 47.1 | **78.0** | **95.9** | **96.1** | 60.1 | 59.7 |
| Ours(w RoPE) | **76.3** | **72.9** | **92.9** | **79.4** | **68.9** | **50.5** | 76.8 | 92.3 | 91.9 | **61.7** | **62.0** |

## 5 Conclusions

We introduced CLIP-IN, a novel two-stage framework designed to enhance CLIP's fine-grained visual perception capabilities. Our approach leverages the unique properties of instruction editing data as a rich source of hard negative image-text pairs, enabling the model to learn subtle visual-semantic distinctions. Furthermore, we integrated semantically rich long captions and adapted the CLIP text encoder with RoPE via knowledge distillation to capture comprehensive contextual information. Our extensive experiments across various zero-shot classification and retrieval benchmarks, fine-grained visual perception tasks, and evaluations on multimodal large language models consistently demonstrate the effectiveness of CLIP-IN in significantly improving performance, particularly in discerning fine-grained visual details. Ablation studies further validated the complementary nature of instruction editing data and long captions in achieving these gains. By effectively harnessing these two distinct data sources, CLIP-IN advances the state-of-the-art in vision-language representation learning and offers a promising direction for building more perceptually accurate and semantically aware multimodal models. Future work could explore the application of CLIP-IN to other vision-language tasks, investigate more sophisticated methods for generating and utilizing long captions, and extend the framework to incorporate even larger and more diverse instruction editing datasets.

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

# A  Comprehensive Evaluation with Diverse Classification Variants

**Dataset.** The IN-1k (ImageNet-1K) [5] benchmark serves as the foundational standard for evaluating image classification models, typically measuring top-1 accuracy on its large-scale validation set. IN-A [13] is a dataset of real-world adversarially filtered images that fool current ImageNet classifiers. IN-V2 [32] introduces a more challenging and modern test set to better reflect current model generalization capabilities, while IN-R [12] (ImageNet-Rendition) assesses robustness to artistic and stylized depictions of object classes, and IN-S [42] (ImageNet-Sketch) evaluates performance on hand-drawn sketches, testing abstraction and shape understanding.

**Results.** Table 9 presents a comprehensive evaluation of our model (Ours) against the baseline models (OpenAI CLIP and Siglip2) across multiple ImageNet variants, including standard classification (IN-1k), adversarial robustness (IN-A), artistic renditions (IN-R), sketch recognition (IN-S), and generalization to new data (IN-V2). Across all backbones, Ours consistently outperforms the corresponding baselines, demonstrating improved generalization and robustness. On the ViT-L backbone, our model achieves higher average accuracy than OpenAI CLIP (+0.2pp on 224 and +0.6pp on 336), with notable gains in IN-R (89.4 vs. 87.8) and IN-S (61.0 vs. 59.6), indicating stronger performance on stylized and abstract visual inputs. When compared to Siglip2 on the SO400m architecture, Ours shows comparable or slightly better results, particularly on IN-R (95.7 vs. 95.3) and IN-S (68.9 vs. 68.1), while maintaining competitive performance on IN-A and IN-V2. The consistent improvements across diverse visual domains suggest that our method enhances the model's ability to generalize beyond standard photographic data, especially in challenging conditions involving artistic variation and abstraction.

Table 9: Performance on ImageNet variants

| Method | Backbone | Res | IN1k | IN-A | IN-R | IN-S | IN-V2 | Avg |
|---|---|---|---|---|---|---|---|---|
| OpenAI CLIP [30] | ViT-L/14 | 224 | 75.5 | 70.8 | 87.8 | 59.6 | 69.8 | 72.7 |
| Ours | ViT-L/14 | 224 | 76.3 | 67.7 | 89.4 | 61.0 | 70.1 | 72.9 |
| OpenAI CLIP [30] | ViT-L/14 | 336 | 76.6 | 77.5 | 89.1 | 61.0 | 70.9 | 75.0 |
| Ours | ViT-L/14 | 336 | 77.0 | 76.0 | 90.6 | 62.9 | 71.3 | 75.6 |
| SigLIP2 [40] | ViT-SO/14 | 224 | 83.2 | 81.5 | 95.3 | 68.1 | 74.3 | 80.5 |
| Ours | ViT-SO/14 | 224 | 83.4 | 81.3 | 95.7 | 68.9 | 74.6 | 80.8 |
| SigLIP2 [40] | ViT-SO/16 | 384 | 84.1 | 83.2 | 96.7 | 70.2 | 76.1 | 82.1 |
| Ours | ViT-SO/16 | 384 | 83.7 | 83.2 | 96.4 | 70.7 | 75.7 | 82.0 |

# B  Additional Ablation Studies

## B.1  Impact of the Symmetric Hard Negative Loss (LHN)

To validate our proposed symmetric hard negative contrastive loss (LHN), we compare it against two other hard negative mining strategies. NegCLIP [53] loss which uses only the image-to-text hard negative term and triplet loss [27] uses separate image-to-text and text-to-image but without the symmetric formulation.

We use our main ViT-L/14@336 model for this ablation. The results in Table 10 confirm the superiority of our symmetric loss design. Our approach consistently and substantially outperforms both alternative strategies across all evaluated benchmarks. Compared to the Triplet loss, our symmetric loss achieves a remarkable +2.8% improvement on ImageNet-1K accuracy and boosts the average short-text retrieval score by 2.0%. The performance gap is even more pronounced when compared to the one-directional NegCLIP loss, where our method shows gains of +3.1% on ImageNet-1K and +2.4% on average retrieval.

## B.2  Impact of Long Caption Data

To ensure the effectiveness of long caption data, we did the following ablation experiment on ViT-L/14@336. Table 11 show that the simpler model does not enhance fine-grained perception; in fact, it achieves an MMVP score of only 20.7. Our full framework, with the long-text component, reaches

Table 10: Ablation studies on Symmetric Hard Negative Loss (L_HN)

| Method | CLS IN-1K Top-1 | Image-Text Retrieval | | | | |
|---|---|---|---|---|---|---|
| | | | Flickr30k | | MSCOCO | |
| | | Avg | T→I | I→T | T→I | I→T |
| Ours (w/ NegCLIP loss) | 73.9 | 70.7 | 77.9 | 90.5 | 48.6 | 65.9 |
| Ours (w/ Triplet loss) | 74.2 | 71.1 | 78.1 | 89.8 | 49.4 | 66.9 |
| Ours (w/ proposed symmetric L_HN) | **77.0** | **73.1** | **79.3** | **93.8** | **51.1** | **68.2** |

a substantially higher score of 26.1. What's more, our full model also demonstrates significantly stronger performance across all retrieval tasks, outperforming the simpler model on both short-text retrieval (73.1 vs. 65.5) and long-text retrieval (76.4 vs. 69.0). In addition, the full framework also maintains a stronger ImageNet-1K classification score (77.0 vs. 76.3). Overall, This experiment confirms that the long-text capability is a necessary component. The synergistic combination of long-caption context and hard-negative training is essential to achieve the reported gains in fine-grained visual perception and overall model performance.

Table 11: Ablation studies on different training data

| Method | CLS IN-1K Top-1 | MMVP | Short Caption Retrieval | | | | | Long Caption Retrieval | | | | |
|---|---|---|---|---|---|---|---|---|---|---|---|---|
| | | | | Flickr | | COCO | | | ShareGPT4V | | DCI | |
| | | | Avg | I→T | T→I | I→T | T→I | Avg | I→T | T→I | I→T | T→I |
| Ours (w/o long captions) | 76.3 | 20.7 | 65.5 | 89.2 | 70.7 | 61.2 | 40.7 | 69.0 | 85.2 | 84.3 | 52.9 | 53.6 |
| Ours (full model) | **77.0** | **26.1** | **73.1** | **93.8** | **79.3** | **68.2** | **51.1** | **76.4** | **93.5** | **91.6** | **58.4** | **61.9** |

## B.3 Impact of Long Caption Data Scale

The identified gap in long caption retrieval performance appears to stem from the disparity in training data scale (18M pairs vs. FG-CLIP's 1B pairs). To address this, we expanded our training dataset from 18M to 30M pairs. The results demonstrate our framework's scalability, as shown in Table 12.

Table 12: Ablation studies on long caption data scale

| Method | CLS IN-1K Top-1 | Short Caption Retrieval | | | | | Long Caption Retrieval | | | | |
|---|---|---|---|---|---|---|---|---|---|---|---|
| | | | Flickr | | COCO | | | ShareGPT4V | | DCI | |
| | | Avg | I→T | T→I | I→T | T→I | Avg | I→T | T→I | I→T | T→I |
| FG-CLIP | 76.1 | 73.6 | 93.7 | **81.5** | 68.9 | 50.9 | 81.8 | 97.4 | **96.8** | 66.7 | 66.1 |
| Ours (18M long caption data) | **77.0** | 73.1 | 93.8 | 79.3 | 68.2 | 51.1 | 76.4 | 93.5 | 91.6 | 58.4 | 61.9 |
| Ours (30M long caption data) | 76.2 | **74.9** | **94.2** | **81.5** | **70.3** | **53.5** | **85.1** | **97.9** | 96.0 | **72.7** | **73.8** |

Our updated model, trained on 30M pairs, achieves a long caption retrieval accuracy of 85.1%, surpassing FG-CLIP's performance of 81.8% by +3.3%. This highlights our framework's capability to excel with significantly less training data (over 30 times less data). Despite the improvements in retrieval performance, our classification results remain similar to those of FG-CLIP, ensuring a balanced overall performance across various metrics.

## C  Details of Compositional Benchmarks

Winoground [38] is a challenging benchmark designed to assess visio-linguistic compositional reasoning. It consists of 400 image-text sets, where each set contains two images and two captions. The captions are minimally different (e.g., differing by a single word related to an object, attribute, or relation) and correspond to distinct images. SugarCrepe [14] is a benchmark specifically curated to evaluate compositional reasoning in vision-language models by generating hard negative examples. It aims to mitigate biases found in other datasets by presenting scenarios that require understanding attribute binding, relations, and verb-centric compositions. It contains over 96,000 image-text pairs across six different types of compositional reasoning challenges. Performance is reported as an overall accuracy score. SPEC [28] is a benchmark designed to test fine-grained spatial and compositional

reasoning. It focuses on evaluating how well models can understand and differentiate images based on precise object arrangements and inter-object relationships described in captions. ARO is a benchmark that evaluates vision-language models' ability to reason about fine-grained visual attributes and relational compositions in images. It challenges models to distinguish subtle differences in object properties and spatial or semantic relationships through carefully designed minimal-pair image-text queries.

## D  Feature Visualization Analysis

In Figure 3, we present a comparative visualization of feature activations across different methods, employing the feature extraction technique proposed by Zhou *et al.* [60]. In the visualization, warmer colors (e.g., yellow) signify higher similarity or relevance to the target concept, while cooler colors (e.g., blue) indicate lower relevance or dissimilarity.

Examining the first set of examples, our proposed method demonstrates a clear ability to precisely localize and identify specific attributes, such as the tie, and distinguish its color as either yellow or red. In contrast, FG-CLIP [50] focuses broadly on the region of the cat without specifically highlighting the tie, thereby lacking fine-grained attribute recognition.

In the second set of examples, our approach continues to exhibit enhanced discriminative capability. It successfully identifies and localizes more nuanced objects or distinct entities within complex scenes, such as a "helmet," a "bear," or a "flamingo" (depending on the specific image query). Conversely, both the original CLIP and FG-CLIP models often produce more diffuse attention maps or fail to accurately pinpoint these specific elements, underscoring the improved fine-grained understanding achieved by our method.

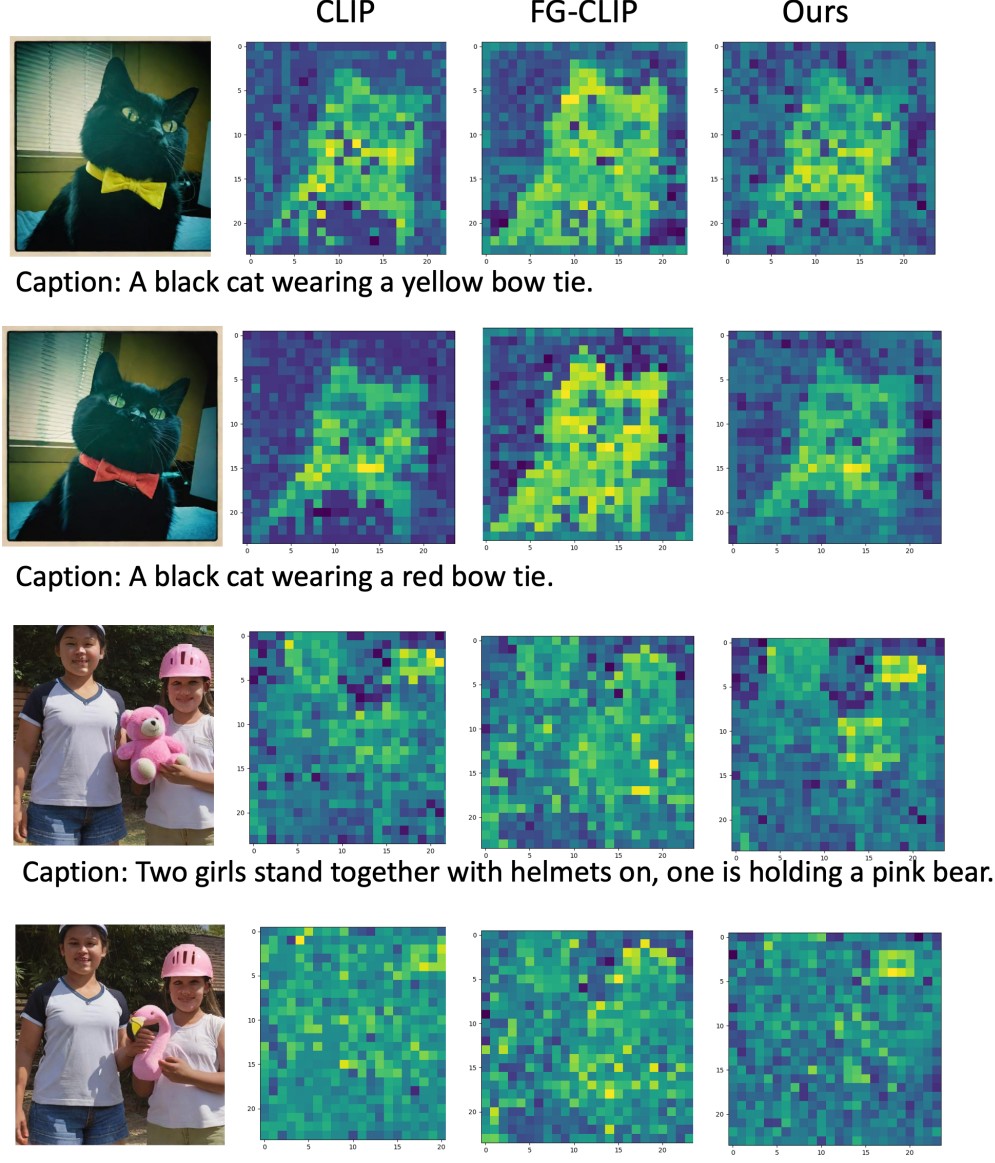

Caption: A black cat wearing a yellow bow tie.

Caption: A black cat wearing a red bow tie.

Caption: Two girls stand together with helmets on, one is holding a pink bear.

Caption: Two girls stand together with helmets on, one is holding a flamingo.

Figure 3: Examples of feature visualization.

# E    Examples of Instruction Editing Hard Image-Text Pairs

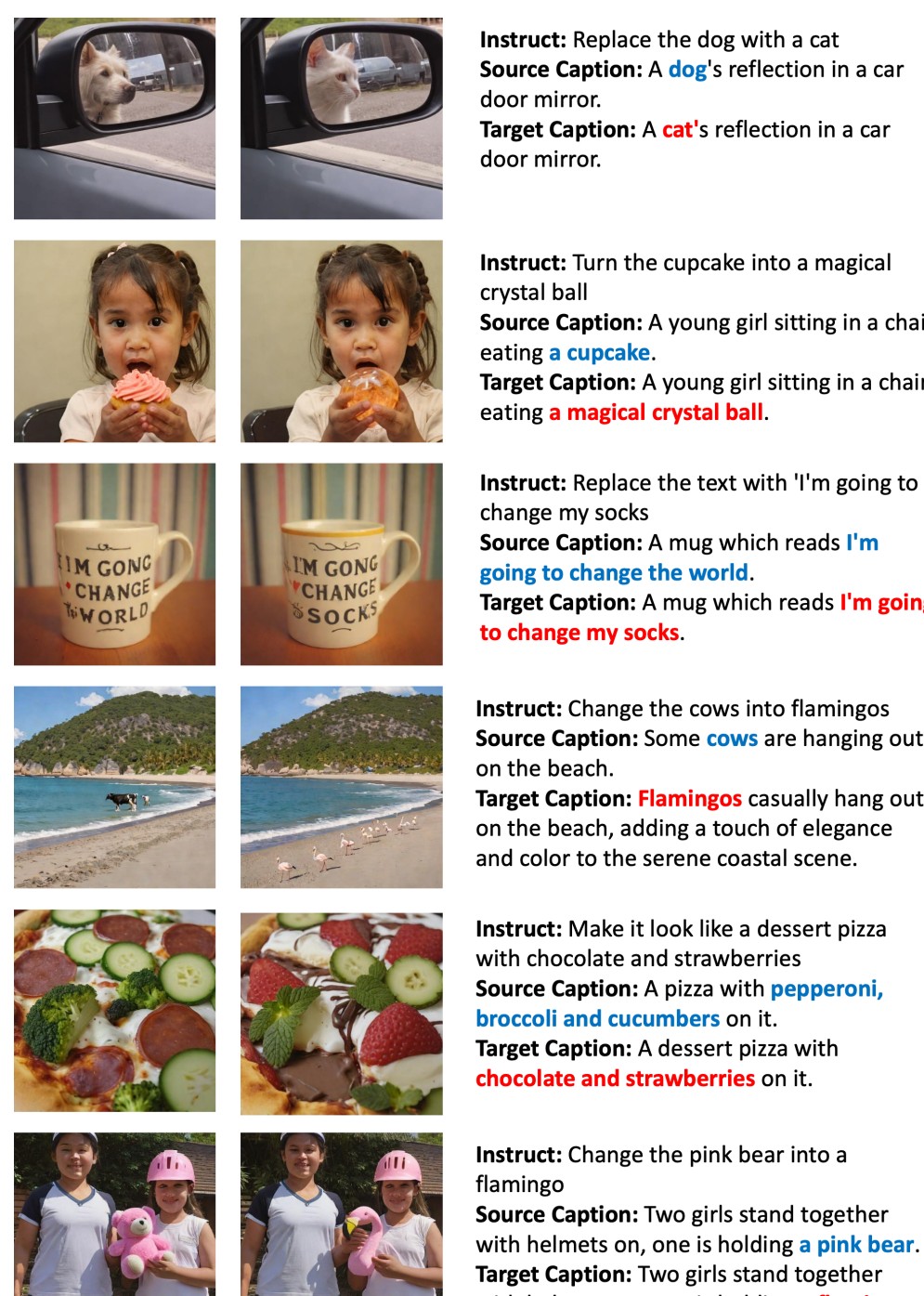

**Instruct:** Replace the dog with a cat
**Source Caption:** A **dog**'s reflection in a car door mirror.
**Target Caption:** A **cat**'s reflection in a car door mirror.

**Instruct:** Turn the cupcake into a magical crystal ball
**Source Caption:** A young girl sitting in a chair eating **a cupcake**.
**Target Caption:** A young girl sitting in a chair eating **a magical crystal ball**.

**Instruct:** Replace the text with 'I'm going to change my socks
**Source Caption:** A mug which reads **I'm going to change the world**.
**Target Caption:** A mug which reads **I'm going to change my socks**.

**Instruct:** Change the cows into flamingos
**Source Caption:** Some **cows** are hanging out on the beach.
**Target Caption:** **Flamingos** casually hang out on the beach, adding a touch of elegance and color to the serene coastal scene.

**Instruct:** Make it look like a dessert pizza with chocolate and strawberries
**Source Caption:** A pizza with **pepperoni, broccoli and cucumbers** on it.
**Target Caption:** A dessert pizza with **chocolate and strawberries** on it.

**Instruct:** Change the pink bear into a flamingo
**Source Caption:** Two girls stand together with helmets on, one is holding **a pink bear**.
**Target Caption:** Two girls stand together with helmets on, one is holding **a flamingo**.

Figure 4: Examples of instruction editing hard image-text pairs.

