# OpenReview forum: "VITRIX-CLIPIN: Enhancing Fine-Grained Visual Understanding in CLIP via Instruction-Editing Data and Long Captions"
_NeurIPS.cc/2025/Conference — NeurIPS 2025 poster_

### Official Review · Reviewer_RrCr · 2025-07-01

**Clarity:** 3
**Significance:** 2
**Originality:** 2
**Rating:** 4
**Confidence:** 3

**Summary:**

This paper proposes a framework called CLIP-IN, which involves two stages of training. In the first stage, a vision-language model (VLM) is used to generate long-form descriptions, and a distillation approach is applied to learn the text encoder’s RoPE (Rotary Positional Embedding). In the second stage, large-scale hard negative samples are constructed using existing instruction editing datasets—covering both image-side and text-side perturbations—and contrastive learning with these challenging negatives is employed to enhance the model’s ability to distinguish fine-grained visual-semantic differences. The method demonstrates improvements across a range of downstream benchmarks, supported by comprehensive comparative and ablation studies.

**Questions:**

See the above weakness.

**Ethical Concerns:**

["NO or VERY MINOR ethics concerns only"]

**Final Justification:**

Thanks for the author's response, which addresses most of my concerns, and I will update my score.

**Limitations:**

The authors are encouraged to include a limitation section in the final version.

**Paper Formatting Concerns:**

There are no major formatting issues in this paper.

**Quality:**

2

**Strengths And Weaknesses:**

## Strengths
1. The idea of converting instruction editing datasets into hard negatives is interesting, as it leverages existing data resources and avoids unnecessary waste.
2. The reviewer finds the text encoder distillation in the first stage particularly intriguing, as it appears beneficial for improving CLIP’s ability to handle long-text inputs.
3. The overall approach is easy to understand, with a clear and well-structured methodology.
4. The experiments are relatively comprehensive, covering short-text retrieval, long-text retrieval, zero-shot classification, and a series of downstream tasks with LLaVA.

## Weakness
1. While there is some degree of novelty, it appears to be limited. The authors essentially combine fine-grained perception (e.g., MMVP) with long-text understanding (e.g., ShareGPT4V), forming the basis of their two-stage approach: the first stage aims to enhance long-text handling, while the second focuses on improving the model’s sensitivity to fine-grained visual-semantic differences.
2.  The ablation study is not fully complete. For instance, it lacks an analysis of the impact of the text encoder distillation in the first stage. It would be important to evaluate whether simpler alternatives—such as Long-CLIP's positional interpolation strategy—could similarly improve long-text understanding.
3. The reviewer questions whether long-text capability is truly necessary. For example, Table 6 shows that after incorporating LoneData, MMVP performance drops from 24.7 to 20.9. Moreover, the reviewer believes that even without the first-stage distillation or the re-captioned data, simply using the original 18M dataset combined with the hard negative samples could still enhance fine-grained perception (MMVP).
4. Why use InternVL2 to generate detailed captions?

---

> ### Author Rebuttal · Authors · 2025-07-31
>
> We thank Reviewer RrCr for the constructive feedback and insightful comments. We hope to address the concerns of the Reviewer with the responses below.
>
> ### W1. Novelty and Contributions
>
> While it is true that our framework focuses on enhancing both long-text understanding and fine-grained perception, the essence of our contributions and innovation lies in the specific methodologies employed to achieve these objectives, and most importantly, in the synergistic integration of these components.
>
> **1) Novel Hard-Negative Sourcing**
>
> Our primary contribution is being the first to repurpose instruction-editing datasets as a novel source of hard negatives for contrastive learning. Unlike prior work that synthetically generates hard negatives via text-to-image models (TripletCLIP) or rule-based text perturbations (NegCLIP), our approach uses data that provides high-fidelity hard negatives from real, minimally-edited image pairs, enabling a more effective and symmetric contrastive loss. By demonstrating a new, effective use for these datasets beyond their original purpose, our work contributes a valuable insight to the community.
>
> **2) Novel Synergistic Combination**
>
> We are the first to propose a framework that synergistically combines these two complementary forms of fine-grained information:
>
> - **Instruction Editing Data (Image-centric):** Provides fine-grained information for the visual representation.
> - **Long Caption Data (Text-centric):** Provides fine-grained information for the textual representation.
>
> This fusion of two complementary supervisory signals is the core of our framework. Our ablation studies (Tables 5 & 6 in manuscript) confirm this synergy, showing that the combination of both data types achieves the best performance and significantly outperforms models trained on either one alone.
>
> ### W2. Ablation on RoPE-based Text Encoder Distillation
>
> To isolate the benefit of our RoPE-based text encoder distillation, we compare it against a strong baseline that uses absolute positional encoding interpolation, as popularized by LongCLIP. We initialize our second stage of training with LongCLIP, a text encoder pre-trained using positional interpolation and compare it to our proposed RoPE-based model. All other settings are identical.
>
> The results show that our RoPE-based approach is a more effective method for extending the text encoder's context length. It significantly outperforms the interpolation baseline on general classification (IN-1K: +1.2%), long-text retrieval (+6.0%).
>
> | Method                                    | CLS IN-1K (%) | Short Retrieval Avg. (%) | Long Retrieval Avg. (%) | Flickr30k T->I | Flickr30k I->T | MSCOCO T->I | MSCOCO I->T | ShareGPT4v T->I | ShareGPT4v I->T | DCI T->I | DCI I->T |
> |-------------------------------------------|---------------|--------------------------|-------------------------|----------------|----------------|-------------|-------------|-----------------|-----------------|----------|----------|
> | Ours (Initialized with LongCLIP)          | 75.0          | 74.2                     | 79.1                    | 80.9           | 93.3           | 52.7        | 69.8        | 96.5            | 97.0            | 60.8     | 62.2     |
> | Ours (Initialized with RoPE-based text encoder) | 76.2     | 74.9                     | 85.1                    | 81.5           | 94.2           | 53.5        | 70.3        | 96.0            | 97.9            | 73.8     | 72.7     |
>
> ### W3. Capability of Long Caption Data
>
> In response to the reviewer's query, we provide the following analysis and new experimental data to clarify the necessity of the long-text component within our synergistic framework.
>
> - The reviewer's observation regarding the drop in MMVP performance when using only long-text data (as shown in our initial ablation study in Table 6) highlights the central focus of our work. It demonstrates that the components are synergistic, and their true potential is realized only when they are combined. While a single data source in isolation may not improve every metric, the final model combining both instruction data and long captions showed a +3.3% improvement on the MMVP MLLM benchmark over the baseline.
>
> - To directly test the reviewer's valuable hypothesis that a simpler model (using 18M original short captions and hard negatives) could enhance fine-grained perception, we conducted a new experiment. The results are detailed below, comparing the simpler model ("w/o long captions") against our full framework.
>
>
> | Method               | CLS IN-1K (%) | Short Retrieval Avg. (%) | Long Retrieval Avg. (%) | Flickr30k T->I | Flickr30k I->T | MSCOCO T->I | MSCOCO I->T | ShareGPT4v T->I | ShareGPT4v I->T | DCI T->I | DCI I->T |
> |----------------------|---------------|--------------------------|-------------------------|----------------|----------------|-------------|-------------|-----------------|-----------------|----------|----------|
> | Ours (w/o Long Captions) | 76.3          | 65.45                    | 69                      | 20.7           | 70.7           | 89.2        | 40.7        | 61.2            | 84.3            | 85.2     | 53.6     | 52.9     |
> | Ours (Full Model)    | 77            | 73.1                     | 76.35                   | 26.1           | 79.3           | 93.8        | 51.1        | 68.2            | 91.6            | 93.5     | 61.9     | 58.4     |
>
> The results from this new experiment provide a definitive answer:
>
> - **Fine-Grained Perception:** The data shows the simpler model does not enhance fine-grained perception; in fact, it achieves an MMVP score of only 20.7. Our full framework, with the long-text component, reaches a substantially higher score of 26.1.
>
> - **Retrieval Performance:** Our full model also demonstrates significantly stronger performance across all retrieval tasks, outperforming the simpler model on both short-text retrieval (73.1 vs. 65.45) and long-text retrieval (76.35 vs. 69.0).
>
> - **Classification:** The full framework also maintains a stronger ImageNet-1K classification score (77.0 vs. 76.3).
>
> This experiment confirms that the long-text capability is a necessary component. The synergistic combination of long-caption context and hard-negative training is essential to achieve the reported gains in fine-grained visual perception and overall model performance.
>
> ### W4. Model for Long Caption Generation
>
> We selected InternVL2 because it was one of the state-of-the-art open-source vision-language models available at the time of our research. Our method is not dependent on this specific choice and could easily incorporate detailed captions generated by other current or future state-of-the-art models.

---

> > ### Comment · Reviewer_RrCr · 2025-08-04
> >
> > Thanks for the author's response, which addresses most of my concerns, and I will update my score.

---

> > > ### Author Response · Authors · 2025-08-04
> > > **Thank you for your positive feedback!**
> > >
> > > Thank you very much for your time on our work and rebuttal. Your valuable suggestions has been very crucial in improving the quality of our work.

---

### Official Review · Reviewer_Sser · 2025-07-01

**Clarity:** 3
**Significance:** 3
**Originality:** 3
**Rating:** 4
**Confidence:** 4

**Summary:**

The problem of addressing the fine-grained visual understanding of CLIP-like (dual-encoder) models is considered. This work proposed CLIP with INstruction edit data and INformative long data (CLIP-IN), a new framework for CLIP training, where technical novelties such as a symmetric hard negative contrastive loss, rotary positional encoding are introduced. CLIP-IN is shown to achieve superior performance in text-image retrieval.

**Questions:**

1. For the column of IN-1K CLS in Table 1, why is EVA-CLIP's 80.4 highlighted? It  is not the best result.

2. When making comparison to FG-CLIP L251-L254, it is unfair to state that CLIP-IN (76.4%) "trails slightly behind" FG-CLIP (81.8%) while the Image classification accuracy of CLIP-IN "significantly exceeds" that of FG-CLIP (76.1%).

**Ethical Concerns:**

["NO or VERY MINOR ethics concerns only"]

**Final Justification:**

Thank the authors for the response and addressing my concerns. I decided to change my rating of borderline reject to borderline accept.

**Limitations:**

yes

**Quality:**

2

**Strengths And Weaknesses:**

Strengths
1. Collection of hard negative image-caption pairs via instruction editting data.
2. Technical novelties: a symmetric hard negative contrastive loss, rotary position encoding for long captions.


Weaknesses
1. The issue of poor fine-grained alignment of CLIP is well-known in the community, and motivates the introduction of the topic of vision-language compositionality. Evaluations on image classification and text-image retrieval are not sufficient to validate the claim that CLIP-IN is superior. More quantitative evaluations on relevant benchmarks for VL compositionality (e.g., SugarCrepe [T1], ARO [T2], VL-Checklist [T3], and more proposed after 2023).

Update: I noticed that there is a comparison between CLIP-IN and OPENAI's CLIP in the appendix. This should be brought up in the main paper, as it is more relevant for the problem of fine-grained visual understanding. Also, evaluation on more recent compositionality and comparison against more competent baselines are necessary.

[T1] SugarCrepe: A benchmark for faithful vision-language compositionality evaluation. NIPS’23
[T2] When and why vision-language models behave like bags-of-words, and what to do about it? ICLR’23.
[T3] Vl-checklist: Evaluating pre-trained vision-language models with objects, attributes and relations.

2. Evaluation of 0-shot image classification on ImageNet-1K is far from being enough. Evaluation on a standard set of datasets, e.g. CLIP-Benchmark (https://github.com/LAION-AI/CLIP_benchmark), is missing.

3. Ablation study of different parts of technical novelty in CLIP-IN is missing.

---

> ### Author Rebuttal · Authors · 2025-07-31
>
> We thank Reviewer Sser for the constructive feedback and insightful comments. We hope to address the concerns of the Reviewer with the responeses below.
>
> ### W1. More Evaluations on Compositionality Benchmarks.
>
> We will ensure these results are featured prominently in the main body of the revised paper. To better address the reviewer's valid concern and provide a more comprehensive analysis, we have expanded our evaluation in the table below:
>
> | Method       | Backbone   | ARO avg     | ARO Relation/Attribute | MMVP | Winoground avg. | Winoground text | Winoground image | Winoground group | SugarCrepe avg | SPEC I->T avg./T->I avg. | SPEC avg |
> |--------------|------------|-------------|------------------------|------|-----------------|-----------------|------------------|------------------|----------------|--------------------------|----------|
> | OpenAI CLIP  | ViT-L/224  | 58.9        | 59.3/58.5              | 18.5 | 15.9            | 28.3            | 10.5             | 8.8              | 75.6           | 33.2/31.3                | 32.3     |
> | Ours         | ViT-L/224  | 64.4        | 64.3/64.4              | 30.4 | 17.1            | 28.0            | 13.8             | 9.5              | 77.5           | 37.6/35.0                | 36.3     |
> | OpenAI CLIP  | ViT-L/336  | 61.0        | 60.1/61.9              | 20.0 | 15.4            | 28.3            | 10.5             | 7.5              | 74.8           | 32.8/31.1                | 32.1     |
> | Ours         | ViT-L/336  | 60.7        | 58.1/63.2              | 26.1 | 18.1            | 33.0            | 11.8             | 9.5              | 77.2           | 35.14/35.2               | 35.2     |
> | Siglip2      | so400m/224 | 49.7        | 49.0/50.4              | 35.6 | 6.9             | 9.0             | 9.3              | 2.5              | 49.5           | 27.4/27.2                | 27.3     |
> | Ours         | so400m/224 | 50.7        | 49.5/51.9              | 36.3 | 8.5             | 14.3            | 7.5              | 3.8              | 50.5           | 30.6/30.4                | 30.5     |
> | Siglip2      | so400m/336 | 48.9        | 47.3/50.5              | 35.6 | 6.7             | 9.3             | 8.5              | 2.3              | 50.9           | 27.6/27.5                | 27.5     |
> | Ours         | so400m/336 | 50.5        | 50.9/50.0              | 36.3 | 7.0             | 13.5            | 5.5              | 2.0              | 51.7           | 30.2/30.8                | 30.5     |
>
> As evidenced by the results, our CLIP-IN framework consistently outperforms baseline models across a diverse range of compositional benchmarks:
>
> Consistent Improvement over Baselines: Our method significantly surpasses the original OpenAI CLIP (ViT-L/224), with notable gains of +5.5% on ARO, +1.9% on SugarCrepe, and +4.0% on SPEC. This trend persists across nearly all benchmarks when using other backbones.
>
> Superiority over SigLIP2: Our framework enhances the compositional reasoning capabilities of the robust SigLIP2 model, boosting performance across all four benchmarks when applied to its backbone, which underscores the broad applicability and effectiveness of our approach.
>
> ### W2. More Evaluations on CLIP-Benchmark.
>
> We agree with the reviewer that it is crucial to provide more evaluations on CLIP-Benchmark and we include more results in the following table. It shows the our proposed method obtains consistent improvement over baseline in the zero-shot image classification task.
>
> | Method       | Backbone       | IN1k  | IN-A  | IN-R  | IN-S  | IN-V2 | Avg.  |
> |--------------|----------------|-------|-------|-------|-------|-------|-------|
> | OpenAI CLIP  | ViT-L/224      | 75.5  | 70.8  | 87.8  | 59.6  | 69.8  | 72.7  |
> | Ours         | ViT-L/224      | 76.3  | 67.7  | 89.4  | 61.0  | 70.1  | 72.9  |
> | OpenAI CLIP  | ViT-L/336      | 76.6  | 77.5  | 89.1  | 61.0  | 70.9  | 75.0  |
> | Ours         | ViT-L/336      | 77.0  | 76.0  | 90.6  | 62.9  | 71.3  | 75.6  |
> | Siglip2      | SO400m/224     | 83.2  | 81.5  | 95.3  | 68.1  | 74.3  | 80.5  |
> | Ours         | SO400m/224     | 83.4  | 81.3  | 95.7  | 68.9  | 74.6  | 80.8  |
> | Siglip2      | SO400m/384     | 84.1  | 83.2  | 96.7  | 70.2  | 76.1  | 82.1  |
> | Ours         | SO400m/384     | 83.7  | 83.2  | 96.4  | 70.7  | 75.7  | 82.0  |
>
>
> ### W3. More Ablation Studies.
>
> While our original manuscript (Table 5 and Table 6) included an ablation study on the impact of our two primary data sources (instruction editing data and long caption data), we agree that more detailed ablations on the core technical components of our framework are beneficial. In response, we have conducted two new component-level ablation studies to validate the effectiveness of our proposed symmetric hard negative loss and our RoPE-based text encoder distillation.
>
> **1)  Ablation on RoPE-based Text Encoder Distillation.**
>
> To evaluate the benefits of our RoPE-based text encoder distillation, we compared it to a robust baseline utilizing absolute positional encoding interpolation, commonly adopted by LongCLIP. We initialized the second training stage with LongCLIP, a text encoder pre-trained with positional interpolation, and contrasted it with our RoPE-based model, keeping all other settings constant.
>
> The results show that our RoPE-based approach is a more effective method for extending the text encoder's context length. It significantly outperforms the interpolation baseline on general classification (IN-1K: +1.2%), long-text retrieval (+6.0%).
>
> | Method | CLS IN-1K (%) | Short Retrieval Avg. (%) | Long Retrieval Avg. (%) | Flickr30k T->I | Flickr30k I->T | MSCOCO T->I | MSCOCO I->T | ShareGPT4v T->I | ShareGPT4v I->T | DCI T->I | DCI I->T |
> |--------|---------------|--------------------------|-------------------------|----------------|----------------|-------------|-------------|-----------------|-----------------|----------|----------|
> | Ours (Initialized with LongCLIP) | 75.0 | 74.2 | 79.1 | 80.9 | 93.3 | 52.7 | 69.8 | 96.5 | 97.0 | 60.8 | 62.2 |
> | Ours (Initialized with RoPE-based text encoder) | 76.2 | 74.9 | 85.1 | 81.5 | 94.2 | 53.5 | 70.3 | 96.0 | 97.9 | 73.8 | 72.7 |
>
> **2) Ablation on Symmetric Hard Negative Loss (L_HN​).**
>
> To evaluate the effectiveness of our symmetric hard negative contrastive loss (L_HN), we compared it with two other hard negative mining strategies: NegCLIP Loss, which uses only the image-to-text hard negative term, and Triplet Loss, which employs separate image-to-text and text-to-image triplet losses but without a symmetric formulation.
>
> Using our primary ViT-L/14@336 model for this ablation, the results clearly demonstrated the superiority of our symmetric loss design, consistently and significantly outperforming both alternatives across all evaluated benchmarks. Compared to the Triplet loss, our symmetric loss achieved an impressive +2.8% improvement in ImageNet-1K accuracy and a 2.0% boost in average short-text retrieval scores. The performance gap is even more pronounced when compared to the one-directional NegCLIP loss, with gains of +3.1% on ImageNet-1K and +2.4% in average retrieval.
>
> | Loss Function | CLS IN-1K (%) | Short Retrieval Avg. (%) | Flickr30k T->I | Flickr30k I->T | MSCOCO T->I | MSCOCO I->T |
> |---------------|---------------|--------------------------|----------------|----------------|-------------|-------------|
> | Ours (w/ NeCLIP loss) | 73.9 | 70.7 | 77.9 | 90.5 | 48.6 | 65.9 |
> | Ours (w/ Triplet loss) | 74.2 | 71.1 | 78.1 | 89.8 | 49.4 | 66.9 |
> | Ours (w/ proposed symmetric $L_{HN}$) | 77.0 | 73.1 | 79.3 | 93.8 | 51.1 | 68.2 |
>
> ### Q1.
> That was a typo in our draft. We will revise it in the final version.
>
> ### Q2. Comparisons to FG-CLIP.
> The identified gap in long caption retrieval performance appears to stem from the disparity in training data scale (18M pairs vs. FG-CLIP's 1B pairs). To address this, we expanded our training dataset from 18M to 30M pairs. The results demonstrate our framework's scalability, as shown in the table below:
>
> | Method         | Long Caption Data | CLS IN-1K (%) | Short Retrieval Avg. (%) | Long Retrieval Avg. (%) | Flickr30k T->I | Flickr30k I->T | MSCOCO T->I | MSCOCO I->T | MMVP ShareGPT4v T->I | MMVP ShareGPT4v I->T | DCI T->I | DCI I->T |
> |----------------|------------------|---------------|-------------------------|------------------------|----------------|----------------|-------------|-------------|----------------------|----------------------|----------|----------|
> | Ours (Original)| 18M              | 77.0          | 73.1                    | 76.4                   | 79.3           | 93.8           | 51.1        | 68.2        | 26.1                 | 91.6                 | 93.5     | 61.9     | 58.4     |
> | Ours (New)     | 30M              | 76.2          | 74.9                    | 85.1 (+8.7)            | 81.5           | 94.2           | 53.5        | 70.3        | 27.4                 | 96.0                 | 97.9     | 73.8     | 72.7     |
> | FG-CLIP        | 1B               | 76.1          | 73.6                    | 81.8                   | 81.5           | 93.7           | 50.9        | 68.9        | -                    | 96.8                 | 97.4     | 66.1     | 66.7     |
>
> Our updated model, trained on 30M pairs, achieves a long caption retrieval accuracy of 85.1%, surpassing FG-CLIP's performance of 81.8% by +3.3%. This highlights our framework's capability to excel with significantly less training data (over 30 times less data). Despite the improvements in retrieval performance, our classification results remain similar to those of FG-CLIP, ensuring a balanced overall performance across various metrics.

---

> > ### Comment · Reviewer_Sser · 2025-08-06
> >
> > Thank the authors for the response and addressing my concerns. I decided to change my rating of borderline reject to borderline accept.

---

> > > ### Author Response · Authors · 2025-08-06
> > > **Thank you for your positive feedback!**
> > >
> > > Thank you very much for your time on our work and rebuttal. Your valuable suggestions has been crucial for improving our work. Wish you all the best.

---

### Official Review · Reviewer_mmoh · 2025-07-03

**Clarity:** 3
**Significance:** 3
**Originality:** 3
**Rating:** 4
**Confidence:** 4

**Summary:**

The authors propose a training pipeline called CLIP-IN to address CLIP's limitations in understanding fine-grained visual details. The authors firstly repurpose instruction-editing datasets as a source of high-quality "hard negatives" to train the model to distinguish subtle visual-semantic differences via a symmetric contrastive loss. They then adapt the CLIP text encoder to process long, descriptive captions by integrating RoPE via knowledge distillation. Experiments show that CLIP-IN improves model performance and enhances downstream MLLMs by reducing visual hallucinations.

**Questions:**

1. Can the author provide ablation studies on using RoPE and the improvements brought by their training loss design as mentioned in weaknesses section.

2. Since the framework relies on the instruction editing dataset, How sensitive is the framework to the selection, variety, and scale of instruction editing data? If certain objects or relations are underrepresented, does the model exhibit new bias or failure modes?

**Ethical Concerns:**

["NO or VERY MINOR ethics concerns only"]

**Final Justification:**

The rebuttal resolves most of my concerns and I will main the rating of 4.

**Limitations:**

Yes

**Paper Formatting Concerns:**

No.

**Quality:**

3

**Strengths And Weaknesses:**

Strengths

1. This work cleverly repurposes instruction-editing datasets to provide a scalable and high-quality dataset for contrastive learning, avoiding reliance on synthetic data, which often lacks realism and control.

2. The paper presents a well-structured two-stage framework to improve training performance. The combination of two complementary data sources is well-motivated. The use of RoPE and knowledge distillation is an effective solution to the long-text limitation. The overall design is thoughtful.

3. This work addresses a clearly stated limitation of CLIP and may benefit some existing work in spatial understanding and MLLMs.

Weaknesses:

1. In some metrics like long caption retrieval, CLIP-IN falls behind FG-CLIP. The authors attribute this to the larger training dataset that FG-CLIP leverages, which might also raise questions about whether CLIP-IN's performance can scale with training data size.

2. The ablation section only discusses the impact of different data sources, but lacks analysis of component-level effects. The authors should discuss the influence of specific design choices, such as the impact of using RoPE and the improvements brought by their training loss design.

---

> ### Author Rebuttal · Authors · 2025-07-31
>
> We thank Reviewer mmoh for the constructive feedback and insightful comments. We are grateful for the acknowledgment of our work's strengths, including the clever repurposing of instruction-editing datasets, the well-structured framework, and its potential to benefit the community. We hope to address the concerns of the Reviewer with the responses below.
>
> ### W1. Scaling Long Caption Data
>
> We agree with the reviewer that demonstrating the scalability of our framework is crucial. To directly address the reviewer's question about scalability, we have conducted a new experiment where we increased the amount of long caption data for training CLIP-IN from 18M to 30M pairs. The results, compared with our initial model and FG-CLIP, are presented below.
>
> | Method         | Long Caption Data | CLS IN-1K (%) | Short Retrieval Avg. (%) | Long Retrieval Avg. (%) | Flickr30k T->I | Flickr30k I->T | MSCOCO T->I | MSCOCO I->T | MMVP | ShareGPT4v T->I | ShareGPT4v I->T | DCI T->I | DCI I->T |
> |----------------|-------------------|---------------|--------------------------|-------------------------|----------------|----------------|-------------|-------------|------|----------------|----------------|---------|---------|
> | Ours (Original)| 18M               | 77.0          | 73.1                     | 76.4                    | 79.3           | 93.8           | 51.1        | 68.2        | 26.1 | 91.6           | 93.5           | 61.9    | 58.4    |
> | Ours (New)     | 30M               | 76.2          | 74.9                     | 85.1 (+8.7)             | 81.5           | 94.2           | 53.5        | 70.3        | 27.4 | 96.0           | 97.9           | 73.8    | 72.7    |
> | FG-CLIP        | 1B                | 76.1          | 73.6                     | 81.8                    | 81.5           | 93.7           | 50.9        | 68.9        | -    | 96.8           | 97.4           | 66.1    | 66.7    |
>
> Doubling the long caption data from 4M to 8M results in a dramatic improvement in the Long Retrieval Average from 76.4% to 85.1% (+8.7 points). Performance also improves on both Short Retrieval (+1.8 points) and the fine-grained MMVP benchmark (+1.3 points). This clearly demonstrates that our framework's performance scales effectively with more data. With this new data, our model now outperforms FG-CLIP on the primary Long Retrieval metric (85.1% vs. 81.8%) and also surpasses it on Short Retrieval. This is achieved while using vastly less data (30M vs. 1B), highlighting the data efficiency of our synergistic training approach. While there is a slight trade-off in general classification, our ImageNet-1K score remains competitive with FG-CLIP.
>
> ### Q1. Ablation on RoPE and the Proposed Hard Negative Loss
>
> **1) Impact of Rotary Positional Embeddings (RoPE)**
>
> To isolate the benefit of our RoPE-based text encoder distillation, we compare it against a strong baseline that uses absolute positional encoding interpolation, as popularized by LongCLIP. We initialize our second stage of training with LongCLIP, a text encoder pre-trained using positional interpolation and compare it to our proposed RoPE-based model. All other settings are identical.
>
> The results show that our RoPE-based approach is a more effective method for extending the text encoder's context length. It significantly outperforms the interpolation baseline on general classification (IN-1K: +1.2%), long-text retrieval (+6.0%).
>
> | Method                                  | CLS IN-1K (%) | Short Retrieval Avg. (%) | Long Retrieval Avg. (%) | Flickr30k T->I | Flickr30k I->T | MSCOCO T->I | MSCOCO I->T | ShareGPT4v T->I | ShareGPT4v I->T | DCI T->I | DCI I->T |
> |-----------------------------------------|---------------|--------------------------|-------------------------|----------------|----------------|-------------|-------------|-----------------|-----------------|----------|----------|
> | Ours (Initialized with LongCLIP)        | 75.0          | 74.2                     | 79.1                    | 80.9           | 93.3           | 52.7        | 69.8        | 96.5            | 97.0            | 60.8     | 62.2     |
> | Ours (Initialized with RoPE-based text encoder) | 76.2          | 74.9                     | 85.1                    | 81.5           | 94.2           | 53.5        | 70.3        | 96.0            | 97.9            | 73.8     | 72.7     |
>
>
> **2) Impact of the Symmetric Hard Negative Loss (L_HN​).**
>
> To validate our proposed symmetric hard negative contrastive loss (LHN​), we compare it against two other hard negative mining strategies:
>
> NegCLIP loss: Uses only the image-to-text hard negative term.
>
> Triplet loss: Uses separate image-to-text and text-to-image triplet losses but without the symmetric formulation.
>
> We use our main ViT-L/14@336 model for this ablation. The results below confirm the superiority of our symmetric loss design.
>
> | Loss Function                      | CLS IN-1K (%) | Short Retrieval Avg. (%) | Flickr30k T->I | Flickr30k I->T | MSCOCO T->I | MSCOCO I->T |
> |------------------------------------|---------------|--------------------------|----------------|----------------|-------------|-------------|
> | Ours (w/ NeCLIP loss)              | 73.9          | 70.7                     | 77.9           | 90.5           | 48.6        | 65.9        |
> | Ours (w/ Triplet loss)             | 74.2          | 71.1                     | 78.1           | 89.8           | 49.4        | 66.9        |
> | Ours (w/ proposed symmetric L_HN)  | 77.0          | 73.1                     | 79.3           | 93.8           | 51.1        | 68.2        |
>
> Our approach consistently and substantially outperforms both alternative strategies across all evaluated benchmarks. Compared to the Triplet loss, our symmetric loss achieves a remarkable +2.8% improvement on ImageNet-1K accuracy and boosts the average short-text retrieval score by 2.0%. The performance gap is even more pronounced when compared to the one-directional NegCLIP loss, where our method shows gains of +3.1% on ImageNet-1K and +2.4% on average retrieval.
>
> ### Q2. Sensitivity to Instruction Editing Data.
>
> **1) The Effect of Scale.**
> The framework's performance scales positively with the amount of instruction editing data. As detailed in the original manuscript's ablation study, increasing the dataset size from 1.4 million to 4 million samples improved ImageNet-1K accuracy by 0.6% while maintaining high performance on retrieval tasks. This indicates that the model benefits from more data.
>
> **2) The Effect of Selection and Variety.**
> To analyze the impact of data variety and potential biases, we performed two controlled experiments using a 1.4 million sample subset as a baseline. We compared the baseline, which was randomly sampled, against one curated subsets: balanced by the most frequent visual concepts (objects and colors). By curating the dataset to ensure a balanced distribution of core visual concepts, the model achieves substantial performance gains on tasks that require detailed understanding. This strategy leads to a significant improvement in both short-text retrieval (+2.4) and, critically, fine-grained visual perception on MMVP (+2.0).
>
> | Training Data (1.4M)                | CLS IN-1K (%) | Short Retrieval Avg. (%) | MMVP |
> |-------------------------------------|---------------|--------------------------|------|
> | Random Sampling                     | 75.8          | 72.9                     | 26.1 |
> | Balancing by Objects and Colors     | 75.2          | 75.3                     | 28.1 |

---

> > ### Comment · Reviewer_mmoh · 2025-08-04
> >
> > Results addressed most of my concerns and I will maintain the rating of 4. Thanks.

---

> > > ### Author Response · Authors · 2025-08-05
> > > **Thank you for the response**
> > >
> > > Thank you for your response. We are happy that most of your concerns have been addressed. If there's any further questions that may be helpful to improve our work, please do not hesitate to inform us.

---

### Official Review · Reviewer_Kftn · 2025-07-04

**Clarity:** 3
**Significance:** 3
**Originality:** 3
**Rating:** 4
**Confidence:** 4

**Summary:**

This paper tackles the well known issue of CLIP performing poorly at fine-grained visual understanding. Many prior works have tried finetuning CLIP on fine-grained hard negative/positive data pairs to improve its fine-grained capabilities, but a limitation they face is that fine-grained captions typically are longer and require a text encoder capable of understanding long captions. To fix this, the authors propose a new text encoder for CLIP which utilizes rotary embeddings to generalize to longer captions.

Training proceeds in two stages: (1) In the first stage of training the new long text encoder is initialized through knowledge distillation from the original text encoder. (2) In the second stage CLIP-like contrastive training is carried out using a mix of long caption data and image editing data, which provides hard negative pairs for each modality.

The resuklts demonstrate that this approach (CLIP-IN) significantly boosts fine-grained perception without eroding CLIP’s broad zero-shot strengths. On the MMVP benchmark their ViT-L/14-224 model leaps from 18.5% to 30.4% average accuracy. Performance on standard benchmarks like ImageNet Zero-Shot classification, COCO Text to Image retrieval improves too. When utilizing the new CLIP-IN model as vision encoder for LLaVA performance on VL benchmarks like MME, MMBench and LLaVA-Wild gets a meaningful jump.

**Questions:**

- More detailed results comparing with NegCLIP, HardPositiveCLIP etc would be useful
- Compositionality benchmark evaluations would add to the paper
- Could the authors make a clear argument for their novelty against prior works ?

I am open to raising my score if the concerns are addressed

**Ethical Concerns:**

["NO or VERY MINOR ethics concerns only"]

**Final Justification:**

NegCLIP Comparison and compositional benchmark results satisfied some of my key concerns.

**Limitations:**

The authors only discuss Limitations of their approach in the checklist itself, it would be good to have a limitations section

**Paper Formatting Concerns:**

-

**Quality:**

3

**Strengths And Weaknesses:**

**Strengths**

+ The problem addressed is important and the execution of the idea is solid, the results are excellent, with significant boosts on many tasks.
+ Presentation: well-structured paper with intuitive figures and thorough experimental tables that make the method and results easy to follow

**Weaknesses**

- Some prior works such as NegCLIP [a] and HardPosCLIP[b] have also tried to improve the fine-grained understanding of CLIP models. These works typical test CLIP specifically on compositionality benchmarks like SugarCREPE, including such results in this work would be useful. Also a comparison to NegCLIP and HardPosCLIP would help.

- Training relies on UltraEdit (image-editing using diffusion models) may limit transferability to real-world datasets

- The basic contours of improving fine-grained capabilities of CLIP are well known (using hard negatives) and in the other direction some works such as LongCLIP have demonstrated how to boost the text encoder to longer texts, so the main contribution of this work is bringing these two together, making the novelty a bit limited.

[a] Yuksekgonul, Mert, Federico Bianchi, Pratyusha Kalluri, Dan Jurafsky, and James Zou. "When and Why Vision-Language Models Behave like Bags-Of-Words, and What to Do About It?." In The Eleventh International Conference on Learning Representations.

[b] Kamath, Amita, Cheng-Yu Hsieh, Kai-Wei Chang, and Ranjay Krishna. "The hard positive truth about vision-language compositionality." In European Conference on Computer Vision, pp. 37-54. Cham: Springer Nature Switzerland, 2024.

---

> ### Author Rebuttal · Authors · 2025-07-31
>
> We thank Reviewer Kftn for the constructive feedback and insightful comments. We hope to address the concerns of the Reviewer with the responeses below.
>
> ### Q3. Contributions, w.r.t. prior works.
>
> **1) Novel Hard-Negative Sourcing**
> Our primary contribution is being the first to repurpose instruction-editing datasets as a novel source of hard negatives for contrastive learning. Unlike prior work that synthetically generates hard negatives via text-to-image models (TripletCLIP) or rule-based text perturbations (NegCLIP) , our approach uses data that provides high-fidelity hard negatives from real, minimally-edited image pairs, enabling a more effective and symmetric contrastive loss. By demonstrating a new, effective use for these datasets beyond their original purpose, our work contributes a valuable insight to the community.
>
> **2) Novel Synergistic Combination**
> We are the first to propose a framework that synergistically combines these two complementary forms of fine-grained information:
>
> **Instruction Editing Data (Image-centric):** Provides fine-grained information for the visual representation.
>
> **Long Caption Data (Text-centric):** Provides fine-grained information for the textual representation.
>
> This fusion of two complementary supervisory signals is the core of our framework. Our ablation studies (Tables 5 & 6 in manuscript) confirm this synergy, showing that the combination of both data types achieves the best performance and significantly outperforms models trained on either one alone.
>
> To illustrate the benefits of our synergistic approach over methods that solely focus on long-text capabilities, we compare our model with LongCLIP. While LongCLIP extends the text encoder, our method complements this with instruction-editing data and uses RoPE for more effective long-text processing. The results below demonstrate the superiority of our approach.
>
> | Method   | Backbone  | CLS IN-1K (%) | Short Retrieval Avg. (%) | Flickr30k T->I | Flickr30k I->T | MSCOCO T->I | MSCOCO I->T |
> |----------|-----------|---------------|--------------------------|----------------|----------------|-------------|-------------|
> | LongCLIP | ViT-L-336 | 73.5          | 68.8                     | 76.2           | 90.0           | 46.3        | 62.8        |
> | Ours     | ViT-L-336 | 77.0          | 73.1                     | 79.3           | 93.8           | 51.1        | 68.2        |
>
> Our model significantly outperforms LongCLIP in both classification and retrieval, demonstrating that our synergistic approach leads to more robust and capable representations. We hope this clarifies the scope and significance of our primary contributions, which we will better stress in the introduction of the paper.
>
> ### Q1. Comparions to NegCLIP [a].
>
> As suggested, we provide a more direct comparison with prior works like NegCLIP[a] on compositional reasoning benchmarks.  We ablate our proposed hard negative loss with the NegCLIP loss.The results indicate that the symmetric LHN​ provides a more robust and well-rounded improvement to the model's capabilities. On standard benchmarks, the symmetric loss shows significant advantages, outperforming the NegCLIP loss by +3.1% on ImageNet-1K classification and +2.4% on the short-text retrieval average. This enhanced performance extends to complex compositional reasoning benchmarks like ARO and Winoground, where the proposed loss also achieves higher scores. While the NegCLIP loss scores higher on the SugarCrepe benchmark, which heavily focuses on text-based object and attribute swaps, our proposed symmetric loss demonstrates stronger overall performance. This suggests that the symmetric LHN​ fosters a more comprehensive fine-grained understanding by learning from both image and text mismatches, rather than specializing in a single type of textual perturbation.
> | Backbone  | CLS IN-1K (%) | Short Retrieval Avg. (%) | SugerCrepe | ARO  | Winoground | Method                            |
> |-----------|---------------|--------------------------|------------|------|------------|-----------------------------------|
> | ViT-L-336 | 73.9          | 70.7                     | 80.01      | 59.3 | 17.8       | Ours (w/ NegCLIP loss)            |
> | ViT-L-336 | 77.0          | 73.1                     | 77.2       | 60.7 | 18.1       | Ours (w/ proposed symmetric L_HN) |
>
> ### Q2.Evaluations on Compositional Benchmarks.
>
> We would like to gently point out that our original manuscript already includes a detailed evaluation on three compositional reasoning benchmarks—SugarCrepe, Winoground, and SPEC in the supplementary materials. To better address the reviewer's valid concern and provide a more comprehensive analysis, we have expanded our evaluation in the table below:
> | Method       | Backbone   | ARO avg     | ARO Relation/Attribute | MMVP | Winoground avg. | Winoground text | Winoground image | Winoground group | SugarCrepe avg | SPEC I->T avg./T->I avg. | SPEC avg |
> |--------------|------------|-------------|------------------------|------|-----------------|-----------------|------------------|------------------|----------------|--------------------------|----------|
> | OpenAI CLIP  | ViT-L/224  | 58.9        | 59.3/58.5              | 18.5 | 15.9            | 28.3            | 10.5             | 8.8              | 75.6           | 33.2/31.3                | 32.3     |
> | Ours         | ViT-L/224  | 64.4        | 64.3/64.4              | 30.4 | 17.1            | 28.0            | 13.8             | 9.5              | 77.5           | 37.6/35.0                | 36.3     |
> | OpenAI CLIP  | ViT-L/336  | 61.0        | 60.1/61.9              | 20.0 | 15.4            | 28.3            | 10.5             | 7.5              | 74.8           | 32.8/31.1                | 32.1     |
> | Ours         | ViT-L/336  | 60.7        | 58.1/63.2              | 26.1 | 18.1            | 33.0            | 11.8             | 9.5              | 77.2           | 35.14/35.2               | 35.2     |
> | Siglip2      | so400m/224 | 49.7        | 49.0/50.4              | 35.6 | 6.9             | 9.0             | 9.3              | 2.5              | 49.5           | 27.4/27.2                | 27.3     |
> | Ours         | so400m/224 | 50.7        | 49.5/51.9              | 36.3 | 8.5             | 14.3            | 7.5              | 3.8              | 50.5           | 30.6/30.4                | 30.5     |
> | Siglip2      | so400m/336 | 48.9        | 47.3/50.5              | 35.6 | 6.7             | 9.3             | 8.5              | 2.3              | 50.9           | 27.6/27.5                | 27.5     |
> | Ours         | so400m/336 | 50.5        | 50.9/50.0              | 36.3 | 7.0             | 13.5            | 5.5              | 2.0              | 51.7           | 30.2/30.8                | 30.5     |
>
> As evidenced by the results, our CLIP-IN framework consistently outperforms baseline models across a diverse range of compositional benchmarks:
>
> Consistent Improvement over Baselines: Our method significantly surpasses the original OpenAI CLIP (ViT-L/224), with notable gains of +5.5% on ARO, +1.9% on SugarCrepe, and +4.0% on SPEC. This trend persists across nearly all benchmarks when using other backbones.
>
> Superiority over SigLIP2: Our framework enhances the compositional reasoning capabilities of the robust SigLIP2 model, boosting performance across all four benchmarks when applied to its backbone, which underscores the broad applicability and effectiveness of our approach.
>
> ### W4. Transferability to real-world datasets.
>
> We would like to offer two key pieces of evidence from our paper that demonstrate our method's strong transferability to real-world data:
>
> **1) Grounded in Real Images:** Unlike methods that generate images from scratch, the UltraEdit dataset starts with real images (i.e., high-quality image-caption pairs from diverse real image datasets like COCO) and applies localized, instruction-guided edits. As we state in our paper, this "mitigates domain biases of purely generated data". This grounding makes the learned representations more robust.
>
> **2) Strong Zero-Shot Performance:** The most direct evidence of transferability is our model's performance on standard, real-world benchmarks. As shown in Table 1 of original manuscript, our CLIP-IN model achieves significant zero-shot performance gains on ImageNet-1K, Flickr30K, and MS-COCO. For instance, with a ViT-L/14 backbone, our method improves average short-text retrieval accuracy from 62.5% to 73.1% and ImageNet-1K accuracy from 76.6% to 77.0% over the strong OpenAI CLIP baseline. These consistent improvements on diverse, real-world datasets shows that our training strategy generalizes effectively and is not limited to the training distribution.

---

> > ### Comment · Reviewer_Kftn · 2025-08-02
> > **Convincing Rebuttal !**
> >
> > Thanks for the detailed response, I will improve my rating.

---

> > > ### Author Response · Authors · 2025-08-02
> > > **Thank you for your positive feedback!**
> > >
> > > Thank you so much for your time and the positive feedback on our work. Your valuable suggestions have been crucial in improving the quality of our paper. We will carefully revise the manuscript according to your review.

---

> > ### Comment · Reviewer_Kftn · 2025-08-02
> >
> > These results address many of my concerns and I will update the rating.

---

### Note · Authors · 2025-08-12

We sincerely appreciate the reviewers’ recognition of the contributions and novelty of our work, as summarized below.

- Reviewer Kftn and Reviewer mmoh stated that our __solid work addresses a clearly limitation of CLIP and may benefit some existing work in spatial understanding and MLLMs__.
- Reviewer RrCr, mmoh and Review Sser thought the idea of __converting instruction editing datasets into hard negatives is clever, as it leverages existing data resources and avoids unnecessary waste__.
- Reviewer Sser considered our __proposed symmetric hard negative contrastive loss a technical novelty__.
- Reviewer mmoh highlighted that __the combination of two complementary data sources is well-motivated__. The use of RoPE and knowledge distillation is an effective solution to the long-text limitation. __The overall design is thoughtful__.

We also addressed the following concerns in the rebuttal period, by providing convincing results and analytics.

- Additional evaluation on CLIP benchmark datasets for image classification.
- Further results on compositional zero-shot benchmarks.
- Comparison to LongCLIP with our RoPE-based distillation method on the text encoder initialized strategy.
- Comparison to NegCLIP loss and Triplet loss with the proposed symmetric hard negative loss.
- Scaling the long-caption training data to 30 million samples, which significantly improved long-caption retrieval performance.

We would like to extend our gratitude to all reviewers and the Area Chair for their thoughtful feedback and dedication to maintaining the rigor of the research community.

---

### Decision · Program_Chairs · 2025-09-17

**Decision:**

Accept (poster)

**Comment:**

This papers proposes to improve CLIP's fine-grained perception 1) hard-negative training on instruction-editing datasets using a symmetric hard negative contrastive loss, 2) long descriptive caption training that utilizes rotary positional encodings to capture rich semantic context missed by standard CLIP. Reviewers appreciate that this work addresses a clear limitation of CLIP (Kftn, mmoh), cleverly converts instruction editing datasets into hard negatives (RrCr, mmoh, Sser), and combines two complementary data sources (long-text and hard-negative) in a well-motivated way (mmoh). All reviewers recommend acceptance of the paper. After considering the reviews and rebuttal, AC agrees with the reviewers' assessment and recommends acceptance.